# RhopH2 and RhopH3 export enables assembly of the RhopH complex on *P. falciparum*-infected erythrocyte membranes

Michał Pasternak [1,2,3], Julie M. J. Verhoef[1,4], Wilson Wong[1,2], Tony Triglia[1], Michael J. Mlodzianoski[1,2], Niall Geoghegan [1,2], Cindy Evelyn [1], Ahmad Z. Wardak [1], Kelly Rogers [1,2] & Alan F. Cowman [1,2 ✉]

RhopH complexes consists of Clag3, RhopH2 and RhopH3 and are essential for growth of *Plasmodium falciparum* inside infected erythrocytes. Proteins are released from rhoptry organelles during merozoite invasion and trafficked to the surface of infected erythrocytes and enable uptake of nutrients. RhopH3, unlike other RhopH proteins, is required for parasite invasion, suggesting some cellular processes RhopH proteins function as single players rather than a complex. We show the RhopH complex has not formed during merozoite invasion. Clag3 is directly released into the host cell cytoplasm, whilst RhopH2 and RhopH3 are released into the nascent parasitophorous vacuole. Export of RhopH2 and RhopH3 from the parasitophorous vacuole into the infected erythrocyte cytoplasm enables assembly of Clag3/ RhopH2/RhopH3 complexes and incorporation into the host cell membrane concomitant with activation of nutrient uptake. This suggests compartmentalisation prevents premature channel assembly before intact complex is assembled at the host cell membrane.

[1] The Walter and Eliza Hall Institute of Medical Research, Parkville, 3052 Victoria, Australia. [2] Department of Medical Biology, The University of Melbourne, Parkville, 3052 Victoria, Australia. [3] Imperial College London, London SW7 2AZ, UK. [4]Present address: Department of Medical Microbiology, Radboudumc Center for Infectious Diseases, Radboud Institute for Molecular Life Sciences, Radboud University Medical Center, PO Box 9101, 6500 HB Nijmegen, Netherlands. ✉email: cowman@wehi.edu.au

*P*lasmodium falciparum, the deadliest of malaria-causing parasites, undergoes rapid cycles of asexual reproduction inside human red blood cells[1–4]. During these cycles, each parasite grows and undergoes nuclear division in a process called schizogony, producing merozoites that upon egress invade new erythrocytes. To sustain its rapid growth, the parasite remodels the host cell[5–7], leading to formation of new permeability pathways (NPP) which enable uptake of nutrients such as simple sugars and nucleotides from the plasma[5,8]. The establishment of NPP happens around 18 h after parasite invasion and precedes the most rapid growth at trophozoite stage and schizogony[9–11]. The exact mechanism of nutrient uptake has not been deciphered thus far but the main role has been ascribed to the *Plasmodium* Surface Anion Channel (PSAC)[10]. The structure and exact composition of the channel remain unknown but three proteins have emerged as the key components: RhopH1, RhopH2 and RhopH3 and therefore the complex is often called the RhopH complex[10,12–18]. The situation is complicated by the fact that RhopH1 is encoded by several paralogues, most notably Clag3.1, Clag3.2 – both expressed in a mutually-exclusive manner[10,19,20], as well as Clag2, Clag8 and Clag9[21]. Clag3 has been shown to span the host cell membrane and believed to be the core component of the PSAC channel[15,22]. The exact roles of RhopH2 and RhopH3 remain obscure but they are required for establishment of the complex at the host cell membrane[16–18] and it has been suggested they are involved in regulation of channel specificity[16]. Recently, the structure of what is believed to be the trafficked form of the RhopH complex has been published[23,24] comprising of a single copy of RhopH2, RhopH3 and Clag3. The structure of the functional channel as well as the membrane insertion mechanism still remain to be solved.

Intriguingly, RhopH proteins are not simply synthesised at the time when PSAC starts to function about 18 h post invasion[9]. Instead, they are synthesised in the schizont stage of the previous cycle[12,25]. They have been hypothesised to form a complex soon after their synthesis upon which they are trafficked into invasion organelles called rhoptries in the forming merozoites[16–18]. During parasite invasion, the rhoptry content is discharged to facilitate invasion and development of the parasitophorous vacuole[26,27]. RhopH3 has been shown to play a role in merozoite invasion although via an unknown mechanism[16,18] however, neither RhopH2 nor Clag3 was required for successful invasion[16,17], raising the possibility that RhopH proteins might also play stand-alone roles as well as functioning as a complex. The RhopH complex is thought to be deposited into the newly-formed parasitophorous vacuole, which separates the parasite from the host cytoplasm[16–18]. It remains unclear how the complex would then reach the host cell cytoplasm but it is likely to occur via the PTEX (*Plasmodium* Translocon of Exported Proteins), which is required for export of proteins to the host cell cytoplasm[28–31]. Indeed, inhibition of PTEX prevented the assembly of RhopH complex on the surface of the red blood cell which led to the conclusion that all RhopH proteins are exported via PTEX[16]. However, one study showed that this is only true for RhopH2 and RhopH3 as the perturbation of PTEX function did not prevent Clag3 from reaching the host cell[31]. How Clag3 reaches the host cell without requiring PTEX is not understood. PTEX-mediated translocation requires protein unfolding, which may be inconsistent with formation of the RhopH complex before translocation. Once in the host cell cytoplasm, it is unknown how the complex forms and reaches the cell surface and whether the incorporation into the host cell membrane can occur spontaneously or is mediated by specialised chaperones. Finally, the mechanism preventing the premature incorporation of the RhopH complex in the parasitophorous vacuole membrane also remains unknown.

Here we show that RhopH2 and RhopH3 are synthesised and trafficked to rhoptries independently. During invasion, Clag3 is released directly into the red blood cell cytoplasm while RhopH2 and RhopH3 remain associated with the membrane of the forming parasitophorous vacuole. Clag3 remains in the host cytoplasm of ring-stage parasites until RhopH2 and RhopH3 are exported from the PV. The complex then associates and is incorporated into the host cell membrane, where it contributes to formation of the nutrient channel.

## Results

**RhopH2 and RhopH3 are synthesised in late trophozoite stage and trafficked to rhoptries independently**. RhopH proteins are thought to be co-expressed at the schizont stage and it has been hypothesised that they form a complex and are trafficked to the newly formed rhoptries consistent with data showing that depletion of either RhopH2 or RhopH3 prevents the trafficking of Clag3 to the rhoptry[16]. However, this would require assembly of the complex in rhoptries on the developing merozoites and disassembly to allowing trafficking through PTEX from the ring stage to the infected erythrocyte. To understand the trafficking of the components of the RhopH complex during *P. falciparum* blood stage development CRISPR-Cas9 technology was used to tag endogenous loci with fluorescent proteins (Supplementary Fig. 1a). This allowed us to express fluorescently tagged RhopH2 and RhopH3 and to follow their localization using super-resolution live imaging. We were not able to obtain fluorescently tagged Clag3.1 or Clag3.2, likely because the presence of a bulky tag interfered with the protein function and may have led to epigenetic switching between these two variants[20,32]. RhopH2 and RhopH3 were first detected in the cytoplasm of late trophozoites (Fig. 1a, b, Supplementary movie 1 and 2). In addition to the diffused signal, which could comprise soluble proteins, ER- or vesicle-inserted protein or a mix of all, well defined loci of RhopH2 and RhopH3 were also apparent. These were likely corresponding to the nascent rhoptries or rhoptry precursors (Fig. 1a, b arrows). During schizogony, the RhopH2 and RhopH3 cytoplasmic signal became weaker and the majority of the fluorescent signal was observed in the well-defined rhoptries (Fig. 1c, d).

Interestingly, RhopH3-mNeonGreen displayed an additional signal in both trophozoite and schizont (Fig. 1a, b, arrows). This signal colocalised with the membrane dye (Fig. 1c, d) and disappeared just before parasite egress (Fig. 1d, Suppl. Movie 2). This suggested the presence of mNeonGreen in the parasitophorous vacuole which likely was not membrane associated but the limited resolution of light microscopy makes it difficult to distinguish. To test whether this fluorescence corresponded to the full-length RhopH3 or the processed C-terminal portion[16], we used CRISPR-Cas9 to create a double-tagged RhopH3 parasite line. The N terminus was tagged with mScarlet while the C terminus was tagged with mNeonGreen. Super-resolution live imaging revealed that mNeonGreen was present both in the rhoptries and in the parasitophorous vacuole while mScarlet was observed only in the rhoptries (Supplementary Fig. 2). This confirmed that the membrane-associated signal consisted of the C terminus of RhopH3. At the same time, rhoptries contained both mScarlet and mNeonGreen which likely corresponded to either full-length protein, processed RhopH3 with both termini remaining associated or a mix of both with the processed N-terminal fragment.

To determine if RhopH2 and RhopH3 are trafficked to rhoptries as a complex as previously suggested[16], we used CRISP-Cas9 to create a parasite line with RhopH2 tagged with mNeonGreen and RhopH3 tagged with mScarlet. In late

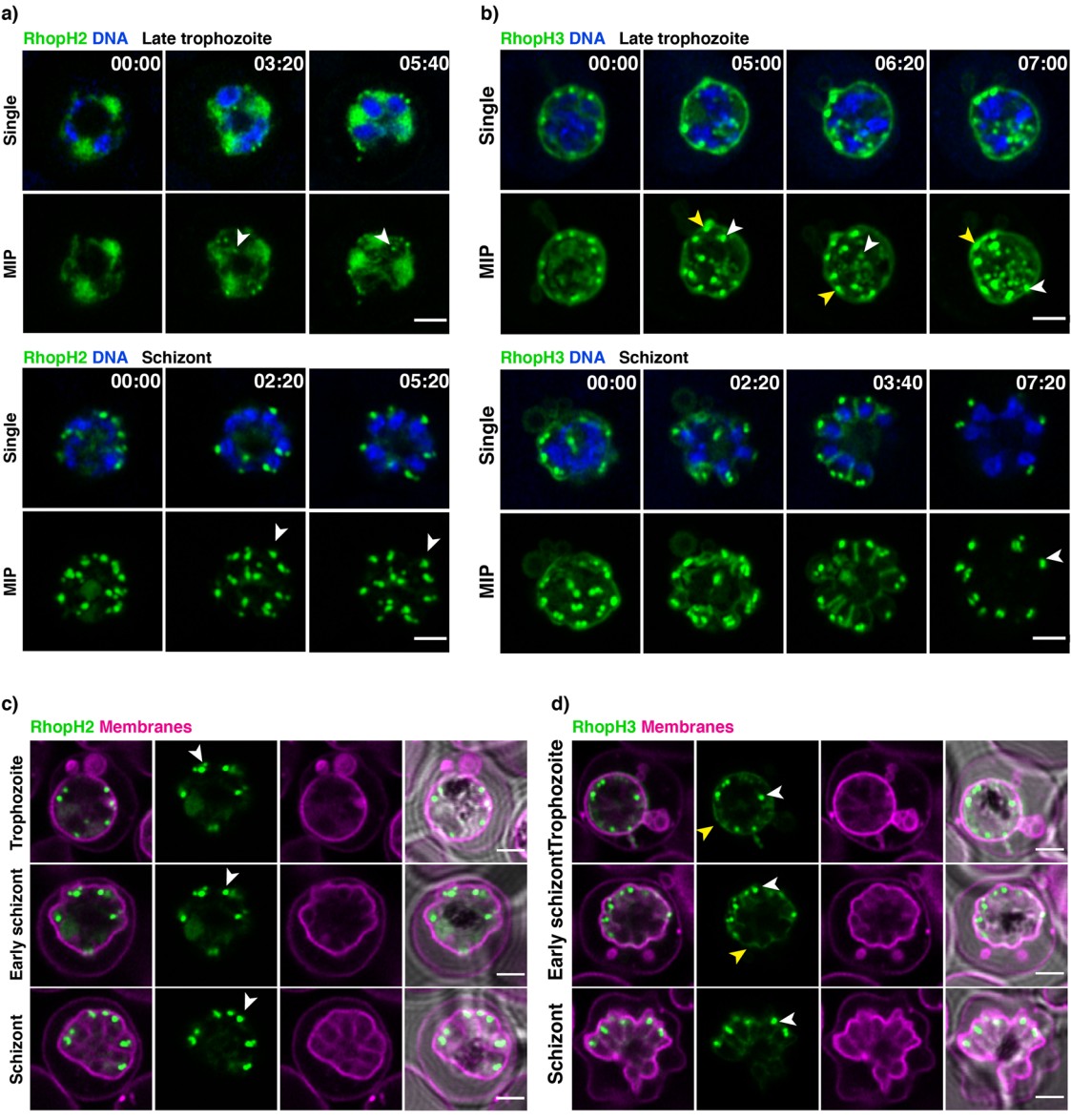

**Fig. 1 Super-resolution live imaging of RhopH2 and RhopH3. a** Live RhopH2-mNeonGreen expressing trophozoites and schizont parasites with SiR-DNA-stained nuclei (blue). mNeonGreen signal accumulation in forming rhoptries - white arrows. **b** Live RhopH3-mNeonGreen expressing trophozoites and schizont parasites with SiR-DNA-stained nuclei (blue). mNeonGreen signal accumulation in forming rhoptries (white arrows) and membrane association (yellow arrows). **c** Live RhopH2-mNeonGreen expressing trophozoites and schizont parasites with membrane dye (purple). **d** Live RhopH3-mNeonGreen expressing trophozoites and schizont parasites with membrane dye (purple). Each panel is a single z section; MIP – maximum intensity projections with a scale bar 2 μm. Time stamps in the upper-right corner represent time points of the overnight time-lapse experiment in the hours: minutes format.

trophozoites, both proteins were detected in the cytoplasm, corresponding to the newly synthesised protein, as well as a defined rhoptry localisation. However, some of these early rhoptries showed only mScarlet or mNeonGreen signal (Supplementary Fig. 3), consistent with these two proteins being trafficked to the newly forming rhoptries or rhoptry precursors separately. This would agree with previous observations that depletion of either RhopH2 or RhopH3 does not affect rhoptry localization of RhopH3[16]. Similarly, RhopH2 was also found in the rhoptries in the absence of RhopH3 although additional diffused signal was observed suggesting that the trafficking to rhoptry might have been less efficient[16].

**During invasion, RhopH2 and RhopH3 are deposited in the parasitophorous vacuole membrane and Clag3 is released in the cytoplasm.** We then decided to follow the fate of RhopH

proteins during the rhoptry discharge at merozoite invasion. For this, we used lattice-light sheet microscopy[33], which allows for unprecedented speed and gentle 3D imaging. Upon invasion, both RhopH2-mNeonGreen and RhopH3-mNeonGreen localised to the newly formed parasitophorous vacuole (Fig. 2a, b, Supplementary movie 3 and 4). To analyse the position of the three proteins as the merozoite interacted with the red cell membrane, parasites were fixed at 1 min 30 s, 10 min, and 30 min post-invasion and proteins of interest were localised using specific antibodies. Localisation of RON4, a component of the tight junction[34–36], was performed to define the stage of merozoite invasion. At early stages of invasion, prior to rhoptry discharge, RON4 localised to the apical tip of the merozoite (Fig. 2c–e, Supplementary Fig. 4). Mid-invasion was marked by a well-defined RON4 ring localisation corresponding to the tight junction formed at the interface between the invading merozoite and

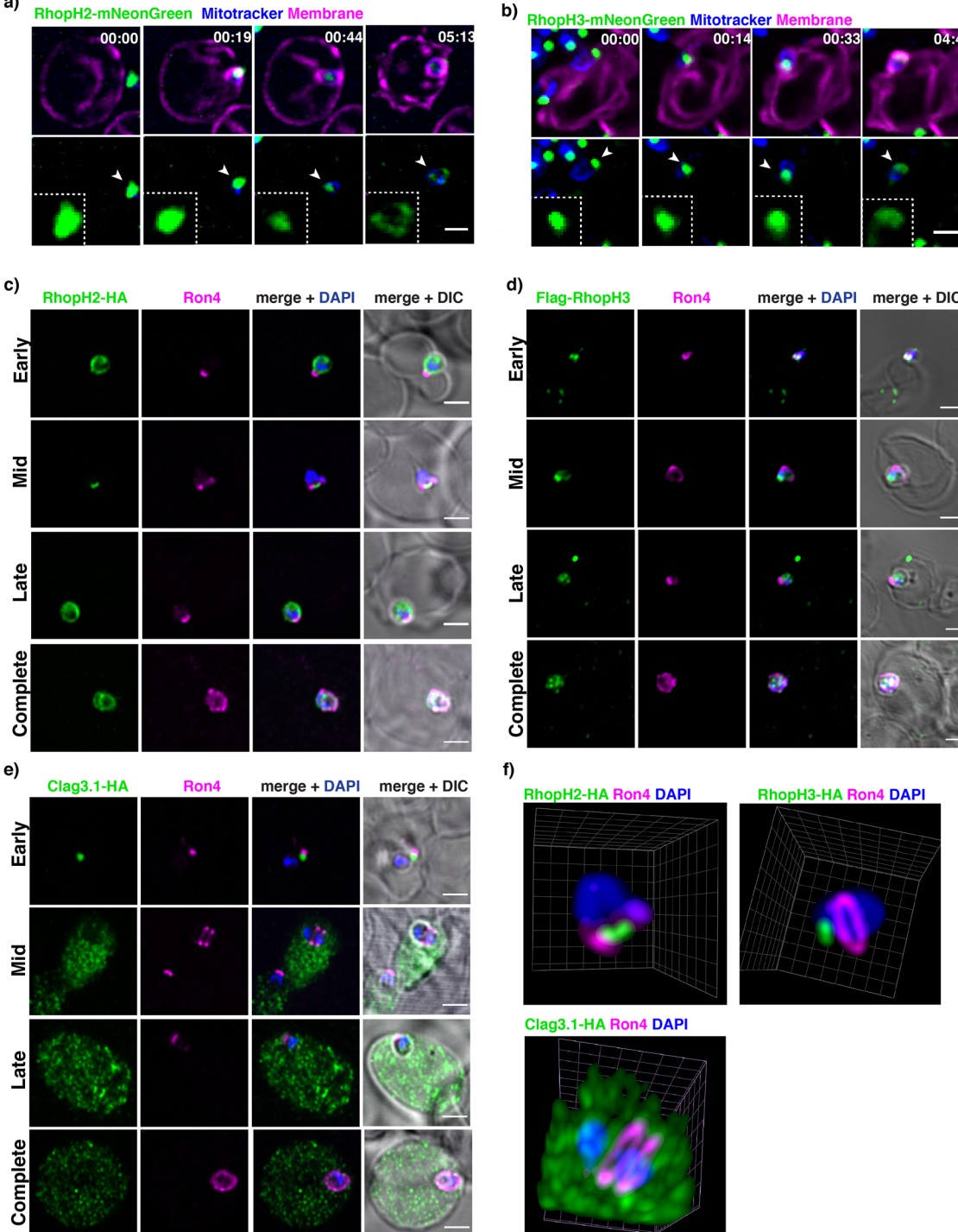

**Fig. 2 Subcellular localisation of RhopH proteins during parasite invasion. a** Live RhopH2-mNeonGreen merozoites invading human erythrocytes imaged using lattice-light sheet microscopy. mNeonGreen signal (white arrows) visible as a bright spot on the merozoite apical end corresponding to RhopH2. The signal became diffused upon successful invasion as the parasitophorous vacuole formed. **b** Live RhopH3-mNeonGreen merozoites invading human erythrocytes imaged using lattice-light sheet microscopy. mNeonGreen signal (white arrows) visible as a bright spot on the merozoite apical end corresponding to RhopH3. The signal became diffused upon successful invasion as the parasitophorous vacuole formed. **c** Localisation of RhopH2-HA and RON4 in fixed merozoites at early, mid, late and complete stages of erythrocyte invasion. RON4 was a marker for the tight junction to assess the stage of invasion. Scale bar 2 μm. **d** Flag-RhopH3 and RON4 in fixed merozoites at early, mid, late and complete stages of erythrocyte invasion. RON4 was a marker for the tight junction to assess the stage of invasion. Scale bar 2 μm. **e** Clag3.1-HA and RON4 in fixed merozoites at early, mid, late and complete stages of erythrocyte invasion. RON4 was a marker for the tight junction to assess the stage of invasion. Scale bar 2 μm. **f** Super-resolution 3D reconstructions of invading merozoites. RON4 labelled in magenta marks the tight junction. HA-tagged Clag3.1, RhopH2, or RhopH3 are labelled in green. Clag3.1 signal appears in the host cytoplasm while RhopH2 and RhopH3 remain concentrated at the apical end of the invading merozoite.

the erythrocyte membrane (Fig. 2c–e, Supplementary Fig. 4). The ring was closing behind the entering merozoite prior to successful completion of invasion (Fig. 2c–e, Supplementary Figure a, Late). Upon successful invasion, RON4 marked the boundaries of the parasitophorous vacuole (Fig. 2c–e, Supplementary Fig. 4, Complete). RhopH2 and RhopH3 localised to the apical tip of the invading merozoite throughout merozoite invasion and were released into the newly formed parasitophorous vacuole upon invasion (Fig. 2 and Supplementary Fig. 4). In contrast, Clag3.1 localised to the apical tip of the invading merozoite at the earliest stages of invasion and was released into the red blood cell cytoplasm as the tight junction was observed (Fig. 2). Clag3.1 remained in the host cell cytoplasm following successful invasion (Fig. 2). These results suggest the RhopH complex is not formed at this stage, in contrast to previous suggestions[16,27]. It was not clear whether RhopH2 and RhopH3 were engulfed in the parasitophorous vacuole in a soluble or membrane-associated form and whether they remained in a complex.

To understand if RhopH2 and RhopH3 associate in a complex we performed invasion assays in the presence of agents inhibiting different stages of merozoite invasion and localization of proteins followed using super-resolution microscopy. To this end, we used R1 peptide, which blocks the Apical Membrane Antigen 1 (AMA1) – RON2 interaction required for formation of the tight junction, thus allowing for the discharge of the rhoptry content but inhibiting the following steps of parasite invasion[34,37]. A monoclonal antibody against the host receptor basigin was also used which blocks binding of the receptor with the parasite ligand Rh5 and prevents rhoptry content discharge and merozoite invasion[37,38].

In DMSO treated controls (Fig. 3a), RhopH2-HA and RhopH3 were both detected in the invading merozoite and the parasitophorous vacuole following successful invasion (10 min time point). In contrast, Clag3.1HA was detected in the cytoplasm of the host erythrocyte during invasion (1 min 30 s time point) and remained there after successful invasion (10 min time point). The addition of R1 peptide prevented the invading merozoites from entering the erythrocyte and forming the parasitophorous vacuole (Fig. 3a, 1 min 30 s time point), resulting in merozoite detachment within 10 min. However, R1 peptide does not prevent discharge of the rhoptry contents and consequently both RhopH2 and RhopH3 were localised to the surface of the red blood cell, forming a gradient starting from the point of merozoite attachment (Fig. 3a, 1 min 30 s time point). As the invading merozoites failed to enter the host cell it detached from the erythrocyte membrane surface leaving behind the RhopH2 and RhopH3 proteins that associated with the erythrocyte membrane that had diffused along the membrane. Interestingly, there was little to no apparent overlap between the RhopH2 and RhopH3 fluorescence signal consistent with our results showing that they are not associated at this stage (Fig. 3a, 1 min 30 s time point). In contrast, Clag3.1 was detected in the erythrocyte cytoplasm even in the presence of R1 peptide (Fig. 3a, R1 peptide, 1 min 30 s time point,) and remained in the cytoplasm upon merozoite detachment (Fig. 3a, R1 peptide, 10 min time point,). In the presence of anti-basigin monoclonal antibody, RhopH2, RhopH3 and Clag3.1 were not discharged from the rhoptries and the fluorescent signal remained within the merozoite boundaries (Fig. 3a, BSG mAb). The R1 peptide and basigin monoclonal antibody inhibited merozoite invasion as quantified by microscopy of thin blood smears (Fig. 3b). These data are consistent with RhopH2, RhopH3 and Clag3 not being in a complex in the merozoite and during invasion. During the merozoite invasion process of the erythrocyte they are deposited into different subcellular compartments of the newly infected cell. Clag3 remains in a soluble form in the erythrocyte cytoplasm

while RhopH2 and RhopH3 associated with the host membrane around the point of invasion and incorporated into the newly-forming parasitophorous vacuole.

**Clag3 is inserted in the host erythrocyte membrane following RhopH2 and RhopH3 export.** The PSAC channel is not functional in early ring stages during *P. falciparum* development but becomes active during the trophozoite stage[38]. To understand the fate of the PSAC components (RhopH2, RhopH3 and Clag3.1) during development parasites were synchronised at ring stage and the subcellular localisation of each protein followed through development. In ring-stage parasites, both RhopH2 and RhopH3 were present predominantly at the parasitophorous vacuole (Fig. 4a). As the ring stage of the parasite developed, both RhopH2 and RhopH3 were exported into the host cell and localised at the erythrocyte membrane (Fig. 4a), where they showed increasing co-localisation (Fig. 4a). In the developing schizonts, the strongest signal was observed in the developing rhoptries as a consequence of the trafficking of newly synthesized RhopH2 and RhopH3 (Fig. 4a). In contrast, in ring-stage parasites, Clag3.1 was observed predominantly in the erythrocyte cytoplasm and it showed no colocalization with RhopH3, which was predominantly within the parasitophorous vacuole (Fig. 4b). Based on the data from invasion experiments, it is apparent that RhopH3 was on the inner surface of the parasitophorous vacuole membrane. However, in trophozoites, Clag3.1 was predominantly in the parasite-infected red cell membrane Fig. 4b, Trophozoite), where it showed colocalization with RhopH3 (Fig. 4b). In developing schizonts, Clag3.1 and RhopH3 colocalized predominantly in the rhoptries and the erythrocyte membrane. These data suggest that incorporation of Clag3.1 into the host cell membrane was concomitant with the export of RhopH2 and RhopH3 from the parsitophorous vacuole and their incorporation into the membrane. This suggests that incorporation of the PSAC complex into the erythrocyte membrane requires the formation of the Clag3, RhopH2 and RhopH3 complex and that arrival of the latter two proteins in the same compartment allows this association and membrane insertion.

To test this possibility and to dissect the protein-protein interactions involved in the formation of the RhopH complex, we performed pull down experiments of RhopH2 from free merozoites, early rings and early trophozoite stages. The co-purified proteins were then analysed by mass-spectrometry (Fig. 4c, Supplementary Data 1). Immuno-blots with the RhopH2-HA pulldown samples revealed that the strongest association between the RhopH complex components was in purified merozoites, when the proteins were packed in rhoptries for delivery into the newly infected cell and in the trophozoite stage, when the complex is assembled in the erythrocyte membrane (Fig. 4d). As expected, the interaction between RhopH2 and Clag3 was reduced in very early rings (Fig. 4d). Despite this, significant Clag3 was detected. This likely results from the association of proteins after solubilisation when membrane compartments and physical barriers are removed allowing them to interact with each other. Surprisingly, the association between RhopH2 and RhopH3 was greatly reduced directly after invasion (Fig. 4d), despite both proteins remaining in the parasitophorous vacuole (Fig. 4a). This was consistent with our observation that RhopH2 and RhopH3 have very little overlap in their subcellular localisation when they are discharged into the host erythrocyte membrane during invasion (Fig. 3). The recent Cryo-EM structure of the soluble form of RhopH complex[23,24] indicates that Clag3 is the core component while RhopH2 and RhopH3 remained associated with the complex via interactions at the opposite ends of Clag3. This agrees with our

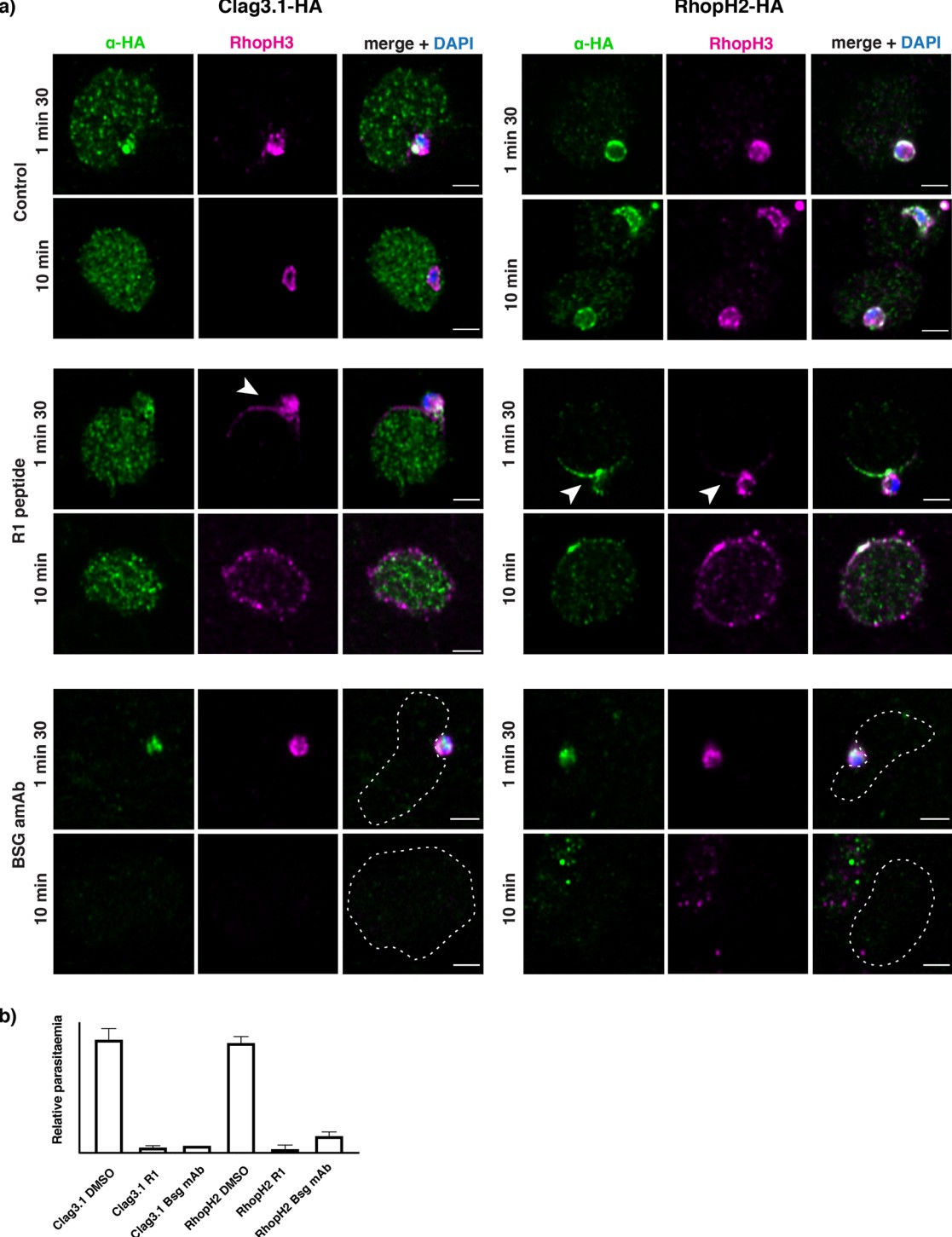

**Fig. 3 RhopH proteins have different cellular localizations during invasion. a** Super-resolution imaging of Clag3.1-HA, RhopH2-HA and RhopH3 (DMSO control). RhopH proteins in the presence of invasion inhibitors with DMSO control (top two panels), R1 peptide (middle two panels) and anti-BSG (bottom two panels). Arrows point to membrane localization of RhopH2 and RhopH3. A dashed line outlines the erythrocyte. Scale bar 2 μm. **b** Invasion efficiency in > 2000 cells per group from two independent experiments for each of the conditions. Error bars show SD.

observation that in the absence of Clag3, the association between RhopH2 and RhopH3 does not occur. We have also attempted to detect protein association using FRET but did not observe any energy transfer in our experimental setup (Supplementary Fig. 7).

We then sought to test if the complex of RhopH2, RhopH3 and Clag3 was able to insert into the erythrocyte membrane spontaneously or if other parasite proteins were required. To this end, we purified the complex from synchronised trophozoites

of a parasite line in which Clag3.1 was tagged with Flag epitopes (Fig. 5a). The RhopH complex was further purified using size-exclusion chromatography. The resulting purified material was visualised by negative-stain electron microscopy (Fig. 5a) and analysed by mass spectrometry (Supplementary Data 2) to show it was homogeneous. This confirmed that the purified complex consisted of Clag3.1, RhopH2 and RhopH3. The purified complex was incubated with erythrocytes and shown to cause lysis

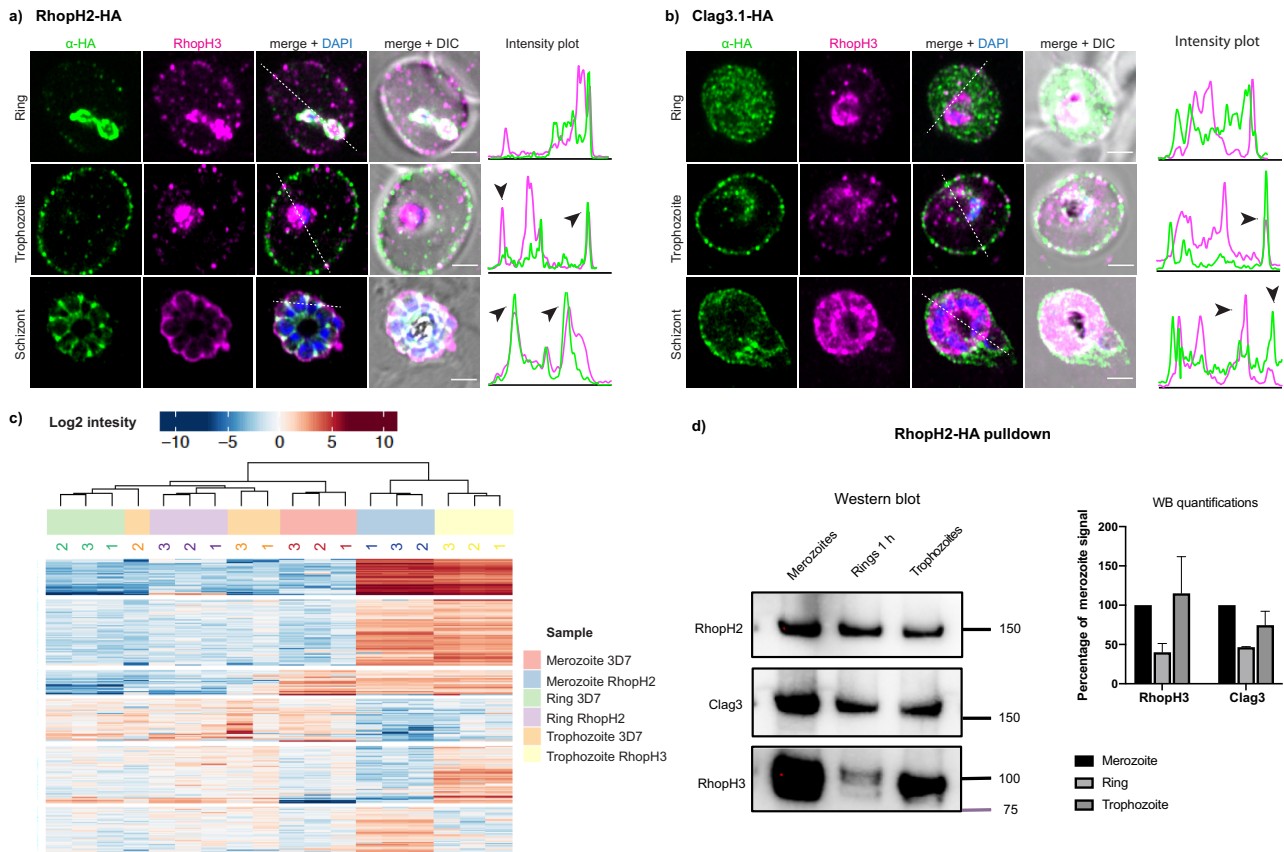

**Fig. 4 RhopH complex reassociates following RhopH2 and RhopH3 export in trophozoite stage. a** Super-resolution images showing colocalization of RhopH2-HA and RhopH3 during parasite development. Intensity plots along the white broken line show lack of colocalization at the ring stage, black arrows point at the points of colocalization on the erythrocyte surface and in the rhoptry. Scale bar 2 μm. **b** Super-resolution images showing colocalization of Clag3.1-HA with RhopH3 during parasite development. Intensity plots are shown along the white broken line that shows lack of colocalization at the ring stage, black arrows point at the points of colocalization on the erythrocyte surface and in the rhoptry. Scale bar 2 μm. **c** Heat map of the mass-spectrometry analysis from RhopH2-HA immunoprecipitation from developmental stages. **d** Immuno-blot (left panel) and quantification (right panel) of RhopH3 and Clag3 from RhopH2-HA immunoprecipitation of developmental stages. Error bars show SD from three biological replicates. Full blots shown (Supplementary Fig. 8).

suggesting it was able to insert into the host cell membrane (Fig. 5b). The lysis leading to the release of haemoglobin was unexpected and we speculate that could be due to the abnormal interaction with the external surface of the red blood cell since physiologically the complex would interact with the inner membrane leaflet on the cytoplasmic side. The insertion attempt from the outer leaflet side, without the presence of supporting cytoskeleton elements, could possibly result in membrane permeabilization and lead to the release of haemoglobin. The lysis mediated by the complex was abolished by heat-treatment prior to mixing with erythrocytes (Fig. 5b). The RhopH complex was incubated with fresh intact erythrocytes as well as erythrocyte ghosts[39] in which the membrane had been permeabilised and the cytoplasmic material extracted to determine if the complex could associate with the erythrocyte membrane. After extensive washing, erythrocytes and ghost cells were pelleted and probed with specific antibodies to RhopH2, RhopH3 or Clag3. The three proteins could be detected in immunoblots suggesting that they were able to interact with the membranes of both intact and ghost red blood cells (Fig. 5c). Together these data show that the RhopH complex can associate with the erythrocyte membrane and suggests that the soluble form can either insert into erythrocyte membranes or remain peripherally associated. It is difficult to distinguish between these two possibilities as previous studies showed that the members of the complex can be extracted

using either detergent or carbonate buffer[16,17]. This insertion/ association with erythrocyte membranes seemed specific, as no association was detected in synthetic liposomes (Supplementary Fig. 6).

## Discussion

According to the previous model, RhopH2, RhopH3 and Clag3 are all synthesised in schizonts and associate together immediately after synthesis[16]. They are then trafficked together to the newly formed rhoptries and released as a complex into the parasitophorous vacuole following merozoite invasion[16,17,25,40]. The complex then translocates through the PTEX translocon to reach the host cell cytoplasm for trafficking to the erythrocyte surface via an unknown mechanism, where they assemble to form the PSAC nutrient channel[16]. In this model, Clag3 and its paralogues constitute the core of the nutrient channel while RhopH2 and RhopH3 are required for Clag3 trafficking into rhoptries and possible involvement in regulation of channel selectivity[10,15,41,42]. Recent FRET studies have also shown that RhopH2 and Clag3 remain in close proximity inside the rhoptries and on red blood cell surface, but there is no data on their association during and immediately after the merozoite invasion[43].

This model does not explain the observation that RhopH3 but not RhopH2 or Clag3, has been implicated in parasite invasion[16,17,40]. There are also mixed reports regarding the role of

## a) RhopH complex purification

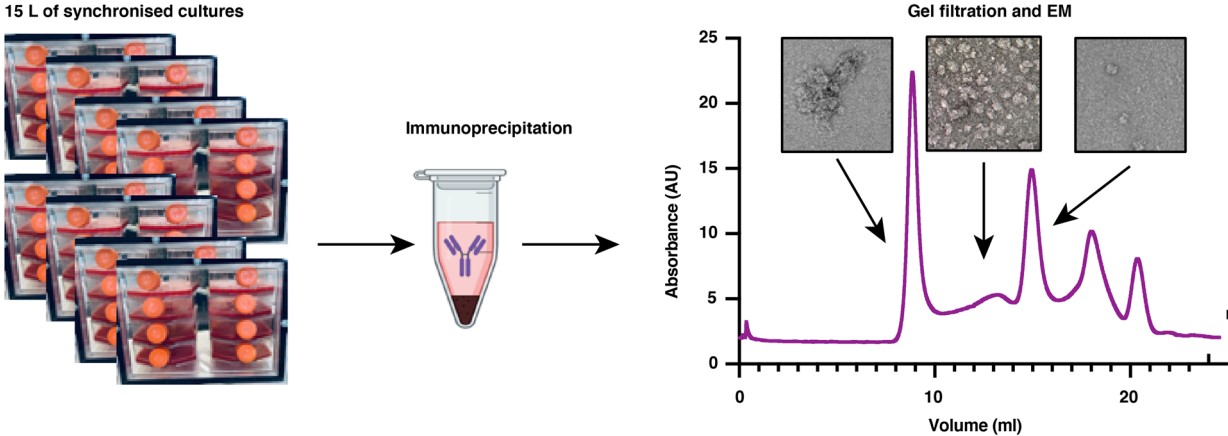

**15 L of synchronised cultures**

Immunoprecipitation

**Gel filtration and EM**

## b) RBC lysis

## c) Membrane association

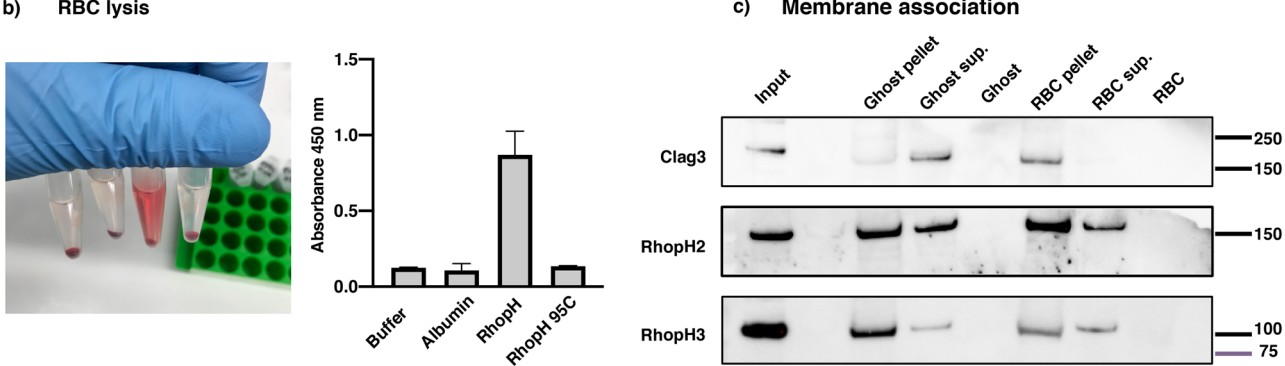

**Fig. 5 Purified RhopH complex spontaneously associates with RBC membranes. a** RhopH complex purified from trophozoites expressing Clag3.1-Flag using an anti-Flag resin. Eluates were further purified by size-exclusion chromatography and analysed by negative-stain electron microscopy and mass spectrometry (left panel). **b** RhopH complex added to human erythrocytes resulted in lysis and haemoglobin release. This lysis activity was abolished if the protein was denatured for 5 min at 95 °C. Quantifications from three independent replicates. **c** Purified RhopH complex can associate with human erythrocytes and ghost membranes. Full blots shown (Supplementary Fig. 8).

the PTEX translocon in the export of RhopH complex from the parasitophorous vacuole. While all previous studies have found that export via the PTEX translocon was required for RhopH2 and RhopH3 to reach the host cell, conflicting results have been reported regarding Clag3[16,31]. Furthermore, since translocation via PTEX requires protein unfolding, the complex would dissociate at this step. Finally, the complex would be trafficked in the parasite-infected erythrocyte cytoplasm and be incorporated into the erythrocyte membrane and how this occurs is unknown. Thus, if all components of the PSAC channel were already associated inside the PV, what mechanism prevents them from incorporation into the parasite membrane or parasitophorous vacuole membrane?

Our results provide data to rewrite the current model of RhopH complex trafficking and assembly and addresses some of the discrepant observations made in previous studies (Fig. 6). Firstly, our results indicate that RhopH2 and RhopH3 are trafficked to the rhoptries independently. This is consistent with previous observations that the absence of either RhopH2 had no impact on the rhoptry localisation of RhopH3[16]. On the other hand, trafficking of Clag3 into rhoptries might be linked to the trafficking of RhopH2 and RhopH3 as Clag3 fails to localise to the rhoptry in the absence of RhopH2 or RhopH3[16]. Also based on FRET studies, RhopH2 and Clag3 remain in close-proximity inside the rhoptry[43] and remain associated in pulldown

experiments. During merozoite invasion, Clag3 is injected directly into the host cell cytoplasm whilst RhopH2 and RhopH3 are directed to the parasitophorous vacuole but do not interact with each other. This is consistent with the Cryo-EM structure of the soluble form of the RhopH complex[23,24], in which RhopH2 and RhopH3 bind directly to Clag3 in the complex but not to each other. Clag3 remains in the host cytoplasm until RhopH2 and RhopH3 are exported via the PTEX translocon, thus allowing the three proteins to associate and incorporate into the erythrocyte membrane. It remains to be determined how the timing of the nutrient channel assembly is regulated by the export of RhopH2 and RhopH3 from the parasitophorous vacuole. Given that discrete foci of RhopH2 or RhopH3 can be detected throughout the ring stage (Fig. 1a, b), the export must happen over time and be regulated, however, the susceptibility to sorbitol at a later stage could reflect the time when a critical mass of the protein has been exported allowing for enough functional channels to be assembled on the RBC surface.

Our results suggest that a role of RhopH2 and RhopH3 is to enable the incorporation of Clag3 into the membrane consistent with Clag3 remaining in the cytoplasm until sufficient RhopH2 and RhopH3 reach the host cytoplasm in late ring stage. This likely explains the conflicting results regarding the role of protein export in nutrient channel assembly. While Clag3 is injected directly into the host cell cytoplasm at merozoite invasion, both

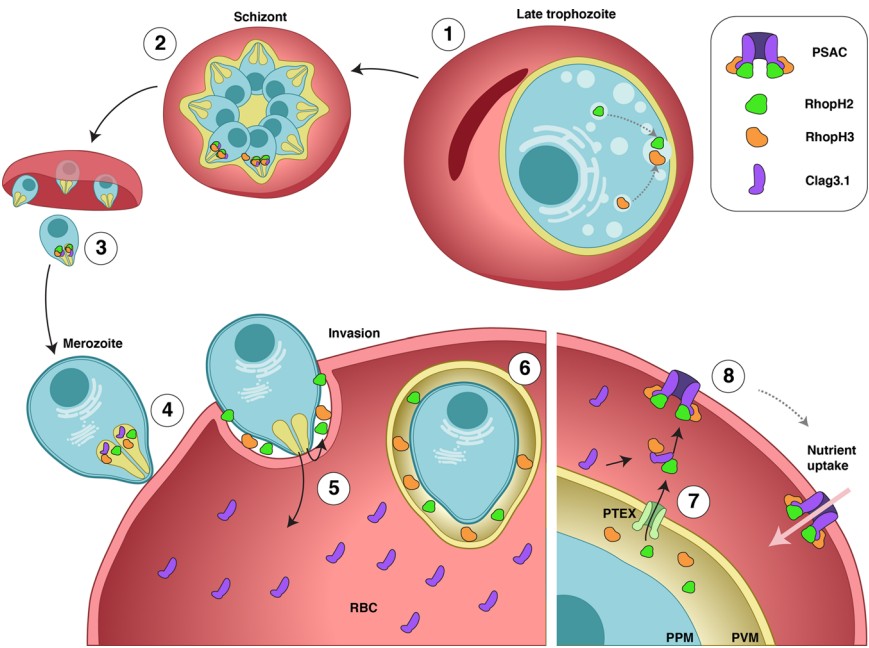

**Fig. 6 Model for RhopH proteins trafficking onto the RBC surface. a** RhopH2 and RhopH3 colocalize in late schizonts (1 and 2). Upon egress (3) free merozoites attach to new red blood cells (4). During invasion, rhoptry content is discharged and RhopH2 and RhopH3 are released into the nascent parasitophorous vacuole while Clag3 is predominantly injected into the host cell cytoplasm (5). Upon successful invasion, RhopH2 and RhopH3 are confined within the parasitophorous vacuole but show very little interaction (6). In trophozoites, RhopH2 and RhopH3 are exported via the PTEX translocon (7). In the host cell cytoplasm, both RhopH2 and RhopH3 associate with Clag3 and the RhopH complex consisting of RhopH2, RhopH3 and Clag3 can spontaneously insert into the erythrocyte membrane, where it forms the nutrient channel (8).

RhopH2 and RhopH3 are exported through the parasitophorous vacuole membrane via PTEX to the host cytoplasm thus allowing the three components to assemble to form the nutrient channel assembly on the erythrocyte surface. This hypothesis is supported by the fact that the complex can spontaneously associate with erythrocyte membranes in the absence of other parasite proteins. The subcellular separation of the RhopH2 and RhopH3 proteins from Clag3 would provide a mechanism preventing premature incorporation of Clag3 and the RhopH complex into membranes (such as the parasitophorous vacuole membrane) until they are all co-located at the red blood cell membrane, where they are required to form the nutrient channel.

Further studies need to be conducted to obtain the structure of the functional PSAC channel to dissect the mechanism by which it permits nutrient import. Thus far only a soluble form of the complex has been obtained[23,24]. Similarly, in this study we only managed to obtain the soluble form despite purifying the complex from synchronised trophozoite culture, when PSAC channel is functional. This suggests that the channel might collapse into a soluble form once extracted from the membranes. Recombinant expression of RhopH proteins is also challenging but if successful, it would allow to overcome the limitations posed by low protein yields obtained from parasite culture. Furthermore, obtaining recombinant proteins would allow for the incorporation studies using various ratios of the complex components to elucidate the insertion mechanism and the role of various Clag3 homologues.

## Methods

**Parasite culture**. Asexual stage 3D7 and CS2 *P. falciparum* parasites were cultured in human O$^+$ erythrocytes at 4% hematocrit in RPMI 1640 medium supplemented with 26 mM HEPES, 50 mg/mL hypoxanthine, 20 mg/mL gentamicin, 0.2% NaHCO$_3$ and 0.5% Albumax II (Gibco). Briefly, cultures were grown in 30 mL petri dishes or 225 cm$^2$ flasks and kept at 37 °C in 1% O$_2$, 5% CO$_2$, 94% N$_2$. Parasite synchronization was done by eliminating stages later than rings with 5% sorbitol.

**Invasion assay**. Synchronised schizonts were purified using Vario MACS CS columns and kept in 1 nM E64 for 4–6 h. Free merozoites were obtained by passing mature schizonts through a 1.2 µm filter, mixed with red blood cells, and incubated at 37 °C while shaking until fixed (at 1 min 30 or 10 min time points) or collected for protein purification.

**Parasite lines**. For creation of pUF1-Cas9G vector pUF1-Cas9[44] was digested with *Xba* I/*Spe* I and religated. The sgRNA cassette was isolated from pL6-eGFP vector using *Aat* II/*Nco* I and inserted into pUF1-Cas9 (DHOD) vector to create pUF1-Cas9G that now encodes both Cas9 and sgRNA expression within the same vector. Guide oligos designed to induce a double-stranded breaks in the corresponding genomic positions were InFusion cloned into pUF1-Cas9G: Clag3.1: TCCCCAAG TTCTACTGCTAA; RhopH2: CTGTTTCATCAACTACCCAT; RhopH3 CCACTT CTTAGATGCTATTG (for C-terminal tagging) or GGCTTATTATAAGCAC CCAC (for N-terminal tagging).

The transgenic parasite lines were made using the CRISPR-cas9 system. The strategy involves the generation of a guide plasmid and a plasmid that replaces the endogenous target gene with a tagged version (the homology-directed repair or HDR plasmid). The HDR plasmid was made in 3 steps, with 5′ and 3′ flanks (500 bp upstream or downstream from the guide sequence) amplified from 3D7 genomic DNA (or synthesised by Genscript) and a codon-optimised target gene sequence (Genscript) downstream of the Cas9 cleavage site fused to the 5′ flank. This was assembled in a modified p1.2 plasmid encoding WR99210 or Blasticidin S (BSD) resistance. 100 µg of linearized HDR plasmid and 100 µg of circular guide plasmid were transfected simultaneously into E64-treated schizonts. Parasites with an integrated drug-resistance cassette were selected and maintained on 2.5 nM WR99210 or 2.5 µg/ml blasticidin S (Sigma). mNeonGreen-tagged parasites were additionally FACS-sorted selecting for the brightest parasites. Genomic integration was confirmed by PCR. The ribbon representation of tagged proteins is shown in Supplementary Fig. 1f.

**Immunofluorescence**. For fixed imaging, intra-erythrocytic parasites were fixed with 4% Paraformaldehyde and 0.01% glutaraldehyde for 30 min, permeabilised with 0.1% TX-100 in HTPBS for 30 min and incubated in blocking solution (2% BSA in PBS) for 1 h. Primary antibodies rat anti-HA (Roche 3F10, 1:300), mouse anti-FLAG (Sigma M2, 1:300), rabbit anti-RhopH2 (Genscript, 1:500), rabbit anti-RhopH3 (Genscript, 1:1000) or rabbit anti-RON4 serum[34] (1:500) were used. Secondary Alexa 488/594 fluorophores were used at 1:1000 dilution. Parasites were mounted on coverslips coated with 1% poly-ethylenimine and mounted with Vectashield containing DAPI (VectorLabs, Australia).

**Airyscan super-resolution microscopy**. Z-stacks of fluorescently labelled infected red blood cells were imaged with Zeiss LSM880 inverted microscope equipped with a Plan Apochromat 63x/1.4 oil objective with 405, 488, 561 and 594 nm excitations and an Airyscan detector. For live microscopy, DNA was labelled with SiR-DNA (Spirochrome), according to manufacturer's instruction), membranes were labelled with Di-4-ANEPPDHQ (Thermo Fisher Scientific) or Bodipy-ceramide (Thermo Fisher Scientific) at 1:500 dilution in RPMI medium supplemented with 0.2% NaHCO$_3$. Live parasite imaging was performed at 37 °C in humidified gas chamber (1% O$_2$ and 5% CO$_2$).

**Lattice light-sheet microscopy**. For all LLSM experiments a custom home-built system was used, constructed as outlined in[33] as per licensed plans kindly provided by Janelia Farm Research campus. Excitation light from either 488 nm, 561 nm, 589 nm or 642 nm diode lasers (MPB Communications) were focused to the back aperture of a 28.6 × 0.7 NA excitation objective (Special optics) via an annular ring of 0.44 inner NA and 0.55 outer NA providing a light sheet 10 μm in length. Fluorescence emission was collected via a 25 × 1.1 NA water dipping objective (Nikon) and detected by either one or two sCMOS cameras (Hamamatsu Orca Flash 4.0 v2). mNeonGreen and Di-4-ANEPPDHQ were excited simultaneously with the 488 nm laser and the resultant fluorescence was split via a 561 nm dichroic (Semrock). SiR-DNA was excited using the 642 nm laser line. mNeonGreen fluorescence was collected through a 525/50 nm filter (Semrock) and Di-4-ANEPPDHQ and SiR-DNA were collected through a 405/488/561/633 multiband filter (Semrock). Mitotracker deep red and Di-4-ANEPPDHQ 525/50 nm and 405/488/561/633 multi-band filter sets were used respectively. All data was acquired in an imaging chamber (Okolabs) set to 36 °C and 5% humidified CO$_2$. For all experiments Point Spread Functions (PSFs) were measured using 200 nm Tetraspeck beads on the surface of a 5 mm coverslip. Data was deskewed and deconvolved using LLSpy, a Python interface for processing of LLSM data. Deconvolution was performed using a Richardson-Lucy algorithm using the PSFs generated for each excitation wavelength.

**Western blot**. Proteins were separated on 4–12% Bis-Tris reducing polyacrylamide gels (Life Technologies) and transferred to nitrocellulose using iBlot (Thermo Fisher Scientific). Membranes were blocked for 1 h with 5% Skim Milk in PBS + 0.1% Tween-20. Monoclonal mouse antibodies against HA (12CA5), FLAG 9H1 and FLAG (M2) were used at 1:1000. Polyclonal Antibodies against RhopH2, RhopH3 and Clag3 were generated in rabbits (GenScript) against the following antigens: RhopH3 T731-Y829, RhopH2 L20-S1378 and Clag3 K1277-H1417. Rabbit polyclonal antibodies were used at 1:500–1:1000 dilution. The following secondary HRP-conjugated antibodies were used: goat a-rat (Southern Biotech, 3030-05), goat α-mouse (Merck Millipore, AP124P), goat a-rabbit (Merck Millipore, AP187P).

**Protein coimmunoprecipitation and mass spectrometry**. Saponin pellets of parasite cultures were lysed in 20 mM HEPES, 150 mM NaCl, 2% DDM, pH 7.2 supplemented with 2x Complete Protease Inhibitors (Roche). Extraction of protein was carried out overnight at 4 °C. Proteins were immunoprecipitated with anti-HA 3F10 (Roche) or FLAG M2 (Sigma) for > 4 h at 4 °C. Beads were thoroughly washed with 20 mM HEPES, 150 mM NaCl, 0.4 mM DDM followed by PBS before proteins were eluted either using a preheated (95 °C) 0.5% SDS for 10 min (HA purification for WB and mass-spec) or 200 μg/ml FLAG peptide (Sigma) in wash buffer (for RhopH complex purification).

**Mass spectrometry**. Mass spectrometry on protein suspension was performed by the Monash Biomedical Proteomics Facility using the following instruments: nano liquid chromatography system: Dionex Ultimate 3000 RSLCnano; mass spectrometer: QExactive HF (Thermo Scientific); analytical column: Acclaim PepMap RSLC (75 μm × 50 cm, nanoViper, C18, 2 μm, 100 Å; Thermo Scientific); Trap column: Acclaim PepMap 100 (100 μm × 2 cm, nanoViper, C18, 5 μm, 100 Å; Thermo Scientific). The raw data files were analyzed using MaxQuant to obtain protein identifications and their respective label-free quantification values using in-house standard parameters. First, contaminant proteins, reverse sequences and proteins identified "only by site" were filtered out. In addition, proteins that have been only identified by a single peptide and proteins that have not been identified consistently have been removed as well. The LFQ data was converted to log2 scale, samples were grouped by conditions and missing values were imputed using the 'Missing not At Random' (MNAR) method, which uses random draws from a left-shifted Gaussian distribution of 1.8 StDev (standard deviation) apart with a width of 0.3. Protein-wise linear models combined with empirical Bayes statistics were used for the differential expression analyses. The 'limma' package from R Bioconductor was used to generate a list of differentially expressed proteins for each pair-wise comparison[45]. A cutoff of the 'adjusted p-value' of 0.05 (Benjamini-Hochberg method) along with a log2 fold change of 1 has been applied to determine regulated proteins in each pairwise comparison.

**Protein expression and purification**. Clag 3.1 fused to a C-terminal FLAG-tag was used as a bait in an affinity step to purify the Clag3.1-FLAG/RhopH2/RhopH3 complex from 10–15 L of transgenic 3D7 *P. falciparum* parasites. Parasite saponin pellet was incubated with lysis buffer (20 mM Tris, 0.15 M NaCl, 2% DDM, pH 7.5) at 4 °C for 1.5 h to extract soluble proteins. Cell debris was removed by ultracentrifugation at 40,000 rpm at 4 °C for 30 min. The soluble fraction containing the Clag3.1-FLAG/RhopH2/RhopH3 complex was incubated with M2 anti-FLAG affinity resin at 4 °C for 60 min. Resins were washed in washing buffer (20 mM Tris, 0.15 M NaCl, 2 mM DDM, pH 7.5) and eluted in elution buffer (20 mM Tris, 0.15 M NaCl, 2 mM DDM, 100 ug/ml FLAG peptide, pH 7.5). Eluted Clag3.1-FLAG/RhopH2/RhopH3 complex was concentrated and loaded onto a size exclusion column (Superose 6 10/300) and eluted in size exclusion buffer (20 mM Tris, 0.15 M NaCl, 2 mM DDM, pH 7.5) to purify the monomeric Clag3.1-FLAG/RhopH2/RhopH3 complex). A total of 10–15 μg of monomeric complex was obtained per purification.

**Negative stain electron microscopy**. Negative-stain electron microscopy was performed at the Bio21 Advanced Microscopy Facility, the University of Melbourne. Three microlitres of purified Clag3.1-FLAG/RhopH2/RhopH3 complex was incubated on glow-discharged holey carbon grids (Quantifoil 1.2/1.3) with a 5 nm continuous carbon support layer for 30 s. Excess sample was removed by blotting on a filter paper, and grids were washed in water before staining in 1% uranyl acetate solution for 30 s. Grids were air-dried and transferred to a FEI TF30 electron microscope operated at 200 kV, with images recorded at a calibrated magnification of 20,500 at defocus values that ranged from 1 to 2 μm. Contrast transfer function parameters were estimated using CTFFIND 4.1.13. Approximately 2000 particles were manually picked and extracted in a box size of 320 Å. Extracted particles were subjected to reference free 2D classification in Relion 3.1 with 50 classes.

**Red blood cell lysis**. For the erythrocyte lysis experiment, 5 μl of purified protein sample (0.4 μg/μl) diluted 1:20 in PBS, bovine albumin was mixed with 100 μl of red blood cells (50% haematocrit) and incubated for 10 min at RT, then 600 μl of PBS was added and samples were incubated at 37 °C overnight. The control samples included bovine albumin at the same concentration as the RhopH complex, purified RhopH complex that had previously been heat-inactivated at 95 °C for 10 min or the buffer in which RhopH complex was stored. Following the overnight incubations, samples were centrifuged at 13,000 g, supernatant has collected the absorbance was measured at 450 nm.

**Membrane incorporation/association**. Ghosts were prepared by repeated washed of red blood cells in 5 mM phosphate (NaH$_2$PO$_4$), pH 8 buffer[39]. Ghost were then resuspended in PBS. For the incorporation experiment, 5 μl of purified protein sample diluted 1:10 in PBS was mixed with 15 μl of ghosts or RBC (10% haematocrit) and incubated for 5 min at RT, then 25 μl of PBS was added and samples were incubated at 37 °C overnight. Following the overnight incubations, samples were centrifuged at 13,000 g, supernatant were collected and pellets washed twice with PBS and collected for immunoblots.

**Antibodies**. In this study, we used the following antibodies (also specified in relevant Methods sections above): Monoclonal antibodies: rat anti-HA (Roche 3F10, Cat.: 11867423001, Lot: 47877600), mouse anti-FLAG (Sigma M2, Cat.: F1804, Lot: SZCD3524), or mouse monoclonal antibodies produced in-house: anti-HA (12CA5), anti-FLAG 9H1. Polyclonal antibodies against RhopH2, RhopH3 and Clag3 were generated in rabbits by GenScript against the following antigens: RhopH3 T731-Y829, RhopH2 L20-S1378 and Clag3 K1277-H1417 and verified using indirect ELISA by the manufacturer. Rabbit anti-RON4 serum was used as published before[34]. The following secondary Alexa 488/594 fluorophores from Life Technologies were used: chicken anti-mouse 594 (Cat.: A21201, Lot: 42099 A), donkey anti-rat 488 (Cat.: A21208, Lot: 2310102), chicken anti-rabbit 594 (Cat.: A21442, Lot: 2110863). The following secondary HRP-conjugated antibodies were used: goat a-rat (Southern Biotech, Cat.: 3030-05, Lot: G2512-M748B), goat α-mouse (Merck Millipore, Cat: AP124P), goat a-rabbit (Merck Millipore, Cat: AP187P).

**Statistics and reproducibility**. Microscopy data on the Lattice Light Sheet was deskewed and deconvolved using LLSpy, a Python interface for processing of LLSM data. Deconvolution was performed using a Richardson-Lucy algorithm using the PSFs generated for each excitation wavelength.

For proteomics the LFQ data was converted to log2 scale, samples were grouped by conditions and missing values were imputed using the 'Missing not At Random' (MNAR) method, which uses random draws from a left-shifted Gaussian distribution of 1.8 StDev (standard deviation) apart with a width of 0.3. Protein-wise linear models combined with empirical Bayes statistics were used for the differential expression analyses. The 'limma' package from R Bioconductor was used to generate a list of differentially expressed proteins for each pair-wise comparison. A cutoff of the 'adjusted p-value' of 0.05 (Benjamini-Hochberg method) along with a log2 fold change of 1 has been applied to determine regulated proteins in each pairwise comparison.

**Reporting summary**. Further information on research design is available in the Nature Research Reporting Summary linked to this article.

## Data availability

The datasets generated during and/or analysed during the current study are available from the corresponding author on reasonable request.

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

## Acknowledgements

We thank the Red Cross Blood Service (Melbourne, Australia) for supply of donor blood for our cell culture. We thank David Steer from Monash Biomedical Proteomics Facility for processing and analysing mass spectrometry samples. The lattice light-sheet referenced in this research was used under license from Howard Hughes Medical Institute, Janelia Research Campus. This work was supported by the National Health and Medical Research Council of Australia (Grants 637406, 1010326, 1049811 and 1057960) and a Victorian State Government Operational Infrastructure Support and Australian Government NHMRC IRIISS. M.P. was supported by an EMBO Long Term Fellowship ALTF 793-2016 and Sir Henry Wellcome Fellowship 206515_Z_17_Z. J.M.J.V. was supported by the NBS travel grant by the Nora Baart Foundation and scholarships from Radboud University and Radboudumc.

## Author contributions

M.P. conceived the study, performed experiments, analysed data, prepared figures and wrote the paper; J.V. performed and analysed experiments and prepared figures; W.W. performed and analysed experiments; C.E. and N.G. performed L.L.S. experiments, T.T. created several parasite lines; M.J.M. performed F.R.E.T. experiments and quantifications; A.Z.W. prepared liposomes. K.R. supervised microscopy experiments, A.F.C. conceived the study, designed and interpreted experiments and wrote the manuscript.

## Competing interests

The authors declare no competing interests.
