## [Peer Review File · Communications Biology]

Reviewers' comments:

Reviewer #1 (Remarks to the Author):

In their manuscript "RhopH2 and RhopH3 export enables assembly of the RhopH complex on the *P. falciparum*-infected erythrocyte membrane" Pasternak and colleagues describe the pathway taken by the members of the RhopH complex from the parasites to the host cell. Using an array of genetically modified parasites that expressed tagged versions of the members of the complex, the authors show that RhopH1, RhopH2 and RhopH3 are unlikely to form a complex inside the parasite, forming a complex only after the RhopH2 and RhopH3 subunits have entered the host cell. These findings would form a very useful addition to the literature, as they provide much needed clarification of the transport of the individual subunits in a field with several contradictory reports and the model presented is a more realistic one than the one proposed previously. The data presented are clear – the microscopy in particular is cutting edge – and the conclusions supported by the data presented. There are, however, several findings that are not directly related to the conclusion that may have important implications beyond the study of the RhopH complex that the authors would do well to address. Furthermore, the language is at times rather vague; using more precise language would make the manuscript clearer. There are also experiments that do not seem to be described in the materials and methods section; these should be described.

Figure 1: It is not clear what the white numbers in the upper right-hand corner of the panels refers to.

It is also not entirely clear to be what the additional value of showing panels a+b and panels c+d is; they appear to convey the same points.

Figure 2: There is little to no membrane-associated RhopH3 in the 10 min R1 peptide sample present. To make the point the protein associates with the membrane, it would be useful to show more images that reveal the membrane association.

Figure 5b: It is interesting that the authors use erythrocyte lysis as a readout for RhopH complex formation as the complex is supposed to form an anion channel that in its native form (when inserted on the inside of the erythrocyte) will not allow for passage of hemoglobin. The authors should provide an explanation of the rationale for this experiment and why the RhopH complex is expected to cause erythrocytes to lyse or release hemoglobin. This experiment is also not described in the materials and methods, making the experiment more difficult to judge.

Figure 5c: Without performing an extraction, membrane incorporation cannot be differentiated from membrane association. As RhopH3 has been shown to bind the erythrocyte membrane by itself (see Sam-Yellowe and Perkins, *Exp. Parasitol.* 73: 161-171 (reference 14)), one can envisage a mechanism by which the complex attaches to the outer membrane of erythrocytes without inserting itself into the membrane. The authors should either perform a carbonate extraction or address the caveat that the results are also compatible with membrane association without insertion of the complex into the membrane.

In several instances the authors use vague language that can lead to some confusion. Line 42-43. "...are deposited at the cell membrane of the infected cell and incorporated into the newly formed parasitophorous vacuole." Just stating that the proteins are released into the nascent PV would be clearer.

Line 69: The different Clag proteins are best described as paralogues, rather than homologues (they may not have the same function; indeed, a different function has been proposed for Clag9 (although that has been disputed subsequently)).

Line 123: The description of RhopH2 and RhopH3 as "cytosolic" is confusing. Do the authors mean that the proteins are free in the cytosol (as could be concluded from this description) or in vesicles in the cytosol (as the model in Figure 6 implies)?

Line 130: 'membrane-associated signal' is somewhat non-specific, perhaps the authors can be more precise about which membrane.

Lines 132 and 235: 'parasitphorous' should be 'parasitophorous'.

Line 223: 'were present predominantly at the parasitophorous vacuole' sounds a bit confusing. One would expect the proteins to be in the parasitophorous vacuole or at the parasitophorous vacuole membrane.

Line 324: "...the three proteins can spontaneously insert.." Presumably this is as a complex, it would be good to state this clearly.

Line 338: Period missing at end of sentence.

Line 378: Did the authors confirm integration by RT-PCR (as written) or by PCR?

Line 465: Was the concentration of DDM used 2% or 2mM (as used in all other buffers)?

Line 489: 'washed' should be 'washing'

Throughout: N-terminus, C-terminus should be N terminus, C terminus.

Line 122: The authors mention that the inability to obtain a Clag3-mNG fusion is likely owing to epigenetic switching and yet they are able to produce a fusion with the Flag tag. This is more compatible with a specific effect of mNG on the Clag3 fusion than epigenetic switching. Although not directly related to this manuscript, the authors may consider whether the ability to add short, loosely folded tags to Clag3, but not large, tightly folded tags, may indicate something about the mechanism through which the protein traverses the PVM.

There are two very important findings that the authors should highlight more and address. First, the argument the RhopH2 and RhopH3 are transported separately is based on the finding that these two proteins are detected in different compartments (Supp Figure 3). This means that two rhoptry markers do not overlap and hence that neither is useful as a rhoptry marker. It would be useful to include a quantification of the number of instances in which the two proteins are not found together. Also, a colocalization of RhopH2 and RhopH3 with RAMA would indicate whether the compartments in which RhopH2 and RhopH3 reside really do represent rhoptries or rhoptry precursors. Also, the lack of colocalization of RhopH2 and RhopH3 is an important argument for the lack of complex formation in the parasites and should be added to the main text.

Second, the authors show that there is a delay in the export of RhopH2 and RhopH3, as these proteins are still in the PVM in the ring stage. This is a significant finding, as it suggests that export through the PTEX can be regulated. RESA, another PTEX cargo, is exported very soon after invasion, so something must be retaining RhopH2 and RhopH3. Costaining with RESA or another early exported protein (REX proteins, perhaps) would show that whereas these exported proteins can be exported, the RhopH proteins are held in the PV. The authors may have identified a new level of regulation of protein export, which should be highlighted more.

Reviewer #2 (Remarks to the Author):

Pasternak et al. use sophisticated imaging technologies with transfected *P. falciparum* lines to study association and trafficking of RhopH proteins involved in formation of the parasite nutrient uptake channel, PSAC. Their findings suggest a revised model of how the proteins form a ternary complex and implicate incorporation into the host membrane to form PSAC as a last step in RhopH trafficking. The work is of a good quality and the interpretations reasonable. However, the revised model would benefit from stronger experimental evidence; discrepancies with prior studies require better discussion also.

1. The most fundamental difference between the proposed model and prior studies is in how the RhopH complex transits from its manufacture to rhoptries and eventually to the cytoplasm of the

next erythrocyte hours after invasion. Prior data suggests assembly of the ternary 1:1:1 complex during translation and a stable association that persists through trafficking to rhoptries and, at least, into the parasitophorous vacuole. In contrast, the authors appear to propose that the three proteins traffic to the rhoptry individually, that they may or may not associate within the rhoptry, and that during invasion that they dissociate to allow Clag3 deposit directly into host cell cytosol while RhopH2 and RhopH3 pass through the parasitophorous vacuole as monomers; the authors' model then presumes assembly (or reassembly) in host cell cytosol and spontaneous insertion into the host membrane. There are a few problems that require discussion in the paper (at a minimum) or new experiments to strengthen the evidence for the new model.

a. The authors acknowledge (Lines 117-118) that the "depletion of either RhopH2 or RhopH3 prevents the trafficking of Clag3 to the rhoptry (ref. #16)." If the proteins transit to the rhoptry individually, Clag3 delivery to rhoptries is not expected to be adversely affected by loss of the other subunits. The authors should discuss this point and list possible explanations to reconcile those findings with their own equally interesting data.

b. lines 293-295: "The biggest caveat of this model is that it does not explain the observation that RhopH3 but not RhopH2 or Clag3, has been implicated in parasite invasion (refs 16, 17, 41). This would require that the complex, believed to be very stable, may dissociate temporarily." This is not necessarily correct. Certainly, a stable exposed domain on RhopH3 could provide the enzymatic activity or receptor-ligand interaction that facilitates invasion without dissociating from other RhopH subunits. One may reasonably speculate that there is a temporary dissociation but "require" seems much too strong.

c. lines 175-177 regarding association of the three subunits within rhoptries: "These results suggest the RhopH complex is either not formed at this stage, in contrast to previous suggestions or that the complex dissociates during invasion." The association of Clag3 and RhopH2 within rhoptries has been supported by FRET studies (see Fig. 6 of PMID: 32900800). As that paper used similar fluorescent tagging of RhopH subunits and live-cell imaging to examine the same questions asked by the authors, it seems important to cite it and discuss points of agreement and disagreement.

d. The co-IP experiments in Fig. 4d suggest a model where the ternary complex between Clag3, RhopH2, and RhopH3 is formed in the rhoptry, with the proteins transiting to this organelle separately. Then, upon egress and invasion, they rapidly dissociate to allow Clag3 secretion into the host cell cytosol very early and RhopH2 and RhopH3 deposition into the parasitophorous vacuole. After these proteins cross the PVM, presumably via PTEX, the three proteins reassociate to allow membrane insertion. The authors' imaging is high-quality, and so, compellingly draws one to this model. At the same time, the RhopH complex structure revealed large surface areas of interaction between Clag3-RhopH2 and Clag3-RhopH3; the ternary complex was also reported to be quite thermostable (ref. 16). There is little or no precedent for such a stable complex to be rapidly disassembled. Given the complex sites of interaction between the member proteins in the RhopH cryo-EM structure, it also seems hard to accept that they would come back together quickly, as invoked by the authors to account for an only modest reduction in Clag3 pulldown by RhopH2 at the ring stage: "Despite this, significant amounts of Clag3 were detected. This may be a result of the association of proteins as they are solubilised and membrane compartments and physical barriers removed allowing them to interact with each other." Can the authors design an experiment to support or refute this concern? At the least, the Discussion section could include the authors' thoughts on this conundrum.

2. Figure 5 and interpretation of hemolysis induced by addition of RhopH complexes to erythrocytes or ghosts.

a. lines 269-271. Although not explicitly stated, the authors appear to interpret hemolysis as incorporation of RhopH and formation of PSAC that increases uninfected erythrocyte/ghost osmotic fragility. I would not expect PSAC induction to produce hemolysis in an overnight PBS incubation as used by the authors because PSAC has a remarkably low Na⁺ permeability (see PMID: 14563534; Na⁺ is the primary osmotically active solute in PBS). I might consider alternate explanations for the observed and still interesting hemolysis.

b. Fig. 5c is labeled "Membrane incorporation". From the Methods, I understand the authors washed the pellet twice with PBS. This is insufficient to remove weakly adherent proteins that are not "incorporated" into the membrane. Association with an ultracentrifugation pellet despite alkaline extraction (e.g. 100 mM NaCO₃ treatment for 30 min as used in ref. 16) is generally required to confidently implicate insertion into membranes.

3. lines 136-142: interpretation of Supplementary Fig. 2 images obtained with super-resolution live imaging. These images are lovely, but I would be more cautious about interpreting where the N- and C- fragments localize based on imaging with two different fluorescent proteins. These proteins have different emission efficiencies so it wouldn't be surprising that both give a signal in the rhoptry (where the protein is concentrated) but give differing signals on the membrane, where there is less protein. To more confidently support such a model, it is generally suggested to swap the two fluorophores (i.e. make a new transfectant that expresses NeonGreen-RhopH3-mScarlet), but this may be prohibitively difficult. This concern also applies to lines 148-150 and Supplementary Fig. 3, where differing intensities of the two reporters are interpreted as implicating separate trafficking of RhopH2 and RhopH3, possibly to separate rhoptries. Also relating to this data, I am skeptical that the c-terminus of RhopH3 associates with membranes (e.g. parasitophorous vacuolar membrane or erythrocyte membrane) without the N-terminus because prior work implicates proteolytic processing near the C-terminus (>10 kDa from end). If correct, this would yield a small C-terminal fragment that lacks a predicted TM. Prior studies suggest that this fragment is rapidly degraded or lost to spent media upon egress (ref. 16, Fig. 6B). N.B. Ref. 16, Fig. 6C and 6F also shows that the vast majority of RhopH3 is proteolytically processed before egress and reinvasion.

Less important points:

lines 134-136: I could not find the details of how a double-tagged RhopH3 parasite line (N-terminal mScarlet and C-terminal mNeonGreen) was made. Sequential transfection vs. a single larger insert? Also, the precise position of the mScarlet relative to the signal sequence is important to describe. To this reviewer's knowledge, this is the first successful N-tagging of any RhopH protein that preserves trafficking and function. As this is not trivial, I would encourage the authors to include a ribbon schematic and highlight this accomplishment. Fig. 2d implies that an N-terminal FLAG tag has also been added. Could the authors provide details for this line, and also RhopH2-HA, Clag3.1HA lines?

Lines 121-123 "We were not able to obtain fluorescently tagged Clag3.1 or Clag3.2, likely due to epigenetic switching between these two variants." But, the authors generated a Clag3.1HA parasite? There are also parasite lines with a single clag3 gene, termed clag3h (PMID: 1815530), which have successfully been used to tag clag3 with fluorescent proteins.

Lines 160-161 "fixed at 1 min, 10 min and 30 min post invasion" implies greater certainty about the timing of invasion and fixation for individual cells in a population than I believe is possible because even tightly synchronized cultures have a distribution of parasite ages. Am I missing something here? The use of RON4 to define progression is elegant and sufficient.

Reviewer #3 (Remarks to the Author):

I think these are excellent findings but not yet definitive enough as the proximity of the postulated associations should and could be demonstrated using FRET. The authors have (some of) the relevant lines already and the choice of mscarlet and Neon Green seems to indicate they were thinking of the approach. The proteins are highly expressed so sensitivity should not be an issue. I'm a little mystified as to why they haven't reported attempting this.

The work is a sophisticated study of the assembly of the heterotrimeric facultative transporter complex consisting of one of the two Clag3 isoforms (RhopH1) and RhopH proteins 3 (H3) and 2 (H2). This manuscript attempts, and to some extent succeeds, to deconvolute the complicated biology of the complex and track how the three members of the complex play merozoite associated roles and how they assemble into a 1:1:1 ratio heterotrimer that is inserted into the host red cell membrane to facilitate nutrient acquisition during parasite vegetative growth in the blood stream. They compare their data largely with that from a recent study of the Desai group (eLife 10:e65282, although the Ho BioRxiv manuscript is also referred to) which followed the fate of Clag3 in detail and refine the model of associations, their timing and nature. The data are not unequivocal and the authors carefully explain the limitations however, with the tools in hand and

generation of others more definitive data could be generated.

CLAG 3 tagging was not achieved possibly due to switching between the pair of analogues – the Desai group deleted one of the pair of genes to allow tagging, an approach that would have improved this study. For example, colocalization via FRET with H3 or H2 in the rhoptries of the segmented merozoite. According to the structure proposed by Desai and colleagues the proximity of the H2 and H3 C termini would be sufficiently close in the assembled complex to allow the appropriate energy transfer. Therefore, a very sensitive and unequivocal demonstration of proximity/assembly of a subunit arrangement that could be anticipated is possible.

The N/CT tagging experiment of H3. What has happened to the processed signal in mature schizonts immediately prior to egress? Contrast Fig1B with Supp Fig 2.

The dual tagging approach ought to allow a FRET based approach which might also discriminate between fragment colocalization and single molecule signals, i.e. there should not be complete overlap in the FRET signal emitted by mscarlet if the processed NT fragment is there on its own

Similarly, a FRET-based approach should show the proximity of H3 and H2 in the rhoptries as described in Supp. Figure 3. FRET signals typically indicate a <10nm distance which would indicate that a complex has or has not been formed. This is of interest as the H3 and H2 signals diverge so markedly upon invasion (and apparently at different rates which is not remarked upon) it begs the question as to why they might be co-complexed in the rhoptries.

This dual tagged line and FRET could also show the relative disposition in Fig 3a b of H2 and H3 during the initial phases of invasion. This would aid accurate conclusions to be drawn resulting from the data in the later images of this figure. Whilst it can't help with CLAG localization unless the specialized line is generated as mentioned above that probably isn't necessary for this final section of the figure as CLAG is so distinct. Nevertheless, an indication of colocalization and possible complex formation in the rhoptries is not beyond this study with a little more work. At the moment, it's a somewhat partial answer.

FRET-based approaches would then allow timing of colocalization and perhaps even ordering of assembly of the complex to be determined as elucidated in Figure 4. Such approaches would allow more definitive language to be employed and avoid the hedging necessarily used from the pull-down approach.

Minor comments

L132 "parasitphorous" spelling

L145 "CRISP" spelling

L235 "parsitophorous" spelling

Supp figure 1. "genmic" spelling in part a

L826. "do" should be "to"

L847 separate and correct "tolate"

Reviewers' comments:

Reviewer #1 (Remarks to the Author):

In their manuscript “RhopH2 and RhopH3 export enables assembly of the RhopH complex on the P. falciparum-infected erythrocyte membrane” Pasternak and colleagues describe the pathway taken by the members of the RhopH complex from the parasites to the host cell. Using an array of genetically modified parasites that expressed tagged versions of the members of the complex, the authors show that RhopH1, RhopH2 and RhopH3 are unlikely to form a complex inside the parasite, forming a complex only after the RhopH2 and RhopH3 subunits have entered the host cell. These findings would form a very useful addition to the literature, as they provide much needed clarification of the transport of the individual subunits in a field with several contradictory reports and the model presented is a more realistic one than the one proposed previously. The data presented are clear – the microscopy in particular is cutting edge – and the conclusions supported by the data presented. There are, however, several findings that are not directly related to the conclusion that may have important implications beyond the study of the RhopH complex that the authors would do well to address. Furthermore, the language is at times rather vague; using more precise language would make the manuscript clearer.

Author response: We would like to thank the reviewer for their kind words and useful suggestions on how to improve the manuscript. We have changed the language to make it more specific and added additional explanation as outlined below.

There are also experiments that do not seem to be described in the materials and methods section; these should be described.

Author response: We thank the reviewer for pointing this out and have now added the relevant sections within the Materials and Methods regarding the protein incorporation into the red blood cells. It reads:

“Red blood cell lysis

For the erythrocyte lysis experiment, 5 μl of purified protein sample (0.4 μg / μl) diluted 1:20 in PBS, bovine albumin was mixed with 100 μl of red blood cells (50% haematocrit) and incubated for 10 min at RT, then 600 μl of PBS was added and samples were incubated at 37°C overnight. The control samples included bovine albumin at the same concentration as the RhopH complex, purified RhopH complex that had previously been heat-inactivated at 95°C for 10 min or the buffer in which RhopH complex was stored. Following the overnight incubations, samples were centrifuged at 13,000 g, supernatant was collected the absorbance was measured at 450 nm.“

Figure 1: It is not clear what the white numbers in the upper right-hand corner of the panels refers to.

Authors response: Figure 1 panels represent different time points of a time-lapse experiments, during which we imaged the parasites expressing fluorescently tagged RhopH2 and RhopH3 overnight. The white numbers mark individual time-points of the time-lapse experiment. We have now clarified that in the legend to Figure 1, which reads:

“Figure 1. Super-resolution live imaging of RhopH2 and RhopH3. (a) Live RhopH2-mNeonGreen expressing trophozoites and schizont parasites with SiR-DNA-stained nuclei (blue). mNeonGreen signal accumulation in forming rhoptries - white arrows. **(b)** Live RhopH3-mNeonGreen expressing trophozoites and schizont parasites with SiR-DNA stained nuclei (blue). mNeonGreen signal accumulation in forming rhoptries (white arrows) and membrane association (yellow arrows). **(c)** Live RhopH2-mNeonGreen expressing trophozoites and schizont parasites with membrane dye (purple). **(d)** Live RhopH3-mNeonGreen expressing trophozoites and schizont parasites with membrane dye (purple). Each panel is a single z section; MIP – maximum intensity projections with a scale bar 2 μm . Time stamps in the upper-right corner represent time points of the overnight time-lapse experiment in the hours: minutes format.”.

It is also not entirely clear to be what the additional value of showing panels a+b and panels c+d is; they appear to convey the same points.

Authors response: We appreciate that the point of showing panels c-d was not made clear. Panels a-b show a distinct localisation of RhopH2 and RhopH3 in a time-lapse live imaging and such discrepancy had never been observed before. Panels a-b come from a long-term time-lapse imaging where the nuclear stain allows for a clear identification for the observed developmental stage. They show that apart from the expected rhoptry signal, RhopH3 displayed additional localisation at the parasite perimeter in late trophozoite and schizonts. Panels c-d show that this additional signal colocalises with a membrane dye and therefore represent membrane associated mNeonGreen from the C terminus of the RhopH3. We have now clarified this in the text:

“Interestingly, RhopH3-mNeonGreen displayed an additional, signal in both trophozoite and schizont (Figure 1a, b, arrows). This signal colocalised with the membrane dye (Figure 1c,d) and disappeared just before parasite egress (Figure 1d, Suppl. Movie 2). This suggested the presence of mNeonGreen in the parasitophorous vacuole and but likely was not membrane associated as the limited resolution of light microscopy makes it difficult to distinguish.”

Figure 2: There is little to no membrane-associated RhopH3 in the 10 min R1 peptide sample present. To make the point the protein associates with the membrane, it would be useful to show more images that reveal the membrane association.

Author response: We believe that the reviewer refers to Figure 3 and we acknowledge the fact that the low brightness of the image made it difficult to observe the RhopH3 signal on the membrane. We have now increased the brightness of the original images and the revised figure is shown below:

Figure 5b: It is interesting that the authors use erythrocyte lysis as a readout for RhopH complex formation as the complex is supposed to form an anion channel that in its native

form (when inserted on the inside of the erythrocyte) will not allow for passage of haemoglobin. The authors should provide an explanation of the rationale for this experiment and why the RhopH complex is expected to cause erythrocytes to lyse or release haemoglobin. This experiment is also not described in the materials and methods, making the experiment more difficult to judge.

Author response: We agree with the reviewer that the release of haemoglobin reflected non-physiological interaction with the RBC membrane. We have added the following explanation into the main text:

“The lysis leading to the release of haemoglobin was unexpected and we speculate that could be due to the abnormal interaction with the external surface of the red blood cell since physiologically the complex would interact with the inner membrane leaflet on the cytoplasmic side. The insertion attempt from the outer leaflet side, without the presence of supporting cytoskeleton elements, could possibly result in membrane permeabilization and lead to the release of haemoglobin.”

In fact, the permeabilization of the RBC membrane has been confirmed with a fluorescent 2-NBD-glucose, where some RBCs appear to have an empty, ghost-like appearance (shown below):

Control

2-NBD-glucose

+RhopH

2-NBD-glucose

As requested by the reviewer, we have also added the relevant method section:

“Red blood cell lysis

For the erythrocyte lysis experiment, 5 µl of purified protein sample (0.4 µg / µl) diluted 1:20 in PBS, bovine albumin was mixed with 100 µl of red blood cells (50% haematocrit) and incubated for 10 min at RT, then 600 µl of PBS was added and samples were incubated at 37°C overnight. The control samples included bovine albumin at the same concentration as the RhopH complex, purified RhopH complex that had previously been heat-inactivated at 95°C for 10 min or the buffer in which RhopH complex was stored. Following the overnight incubations, samples were centrifuged at 13,000 g, supernatant was collected the absorbance was measured at 450 nm.“

Figure 5c: Without performing an extraction, membrane incorporation cannot be differentiated from membrane association. As RhopH3 has been shown to bind the erythrocyte membrane by itself (see Sam-Yellowe and Perkins, *Exp. Parasitol.* 73: 161-171 (reference 14), one can envisage a mechanism by which the complex attaches to the outer membrane of erythrocytes without inserting itself into the membrane. The authors should either perform a carbonate extraction or address the caveat that the results are also compatible with membrane association without insertion of the complex into the membrane.

Author response: The reviewer makes a valid point. Unfortunately, based on previous experiments, which show that the RhopH complex can be extracted by either detergent or bicarbonate buffer, it is difficult to distinguish between the full membrane incorporation vs membrane association of the RhopH complex. Ito et al, 2017, showed that all of RhopH3 and a substantial proportion of RhopH2 and Clag3 can be extracted using bicarbonate buffer. Counihan et al, 2017 managed to extract majority of RhopH2 using bicarbonate buffer as evident from very little protein left in the pellet. It has also been shown that bicarbonate buffer can be used to extract integral proteins with moderately hydrophobic transmembrane domains (Kim H, et al, 2015 Protein Sci. 2015, PMC4815233). Together, this makes it very difficult to use this method to assess whether the in vitro incorporation resulted in a membrane integration or association.

Counihan et al, 2017 Figure 2

Daisuke et al, 2017, Figure 1

We also believe that the protein presence in the pellets washed with PBS was not due to unspecific sticking of the protein to membranes as we repeated the same experiment with synthetic liposomes with the lipid composition similar to that of RBC membrane, and observed very no RhopH3 or Clag3 association with the liposome pellets (shown below).

Therefore, we believe the signal described in Figure 5 was specific although we can't distinguish whether the complex was fully integrated or membrane associated. We therefore follow the reviewer's suggestion and discuss this caveat in the text:

“Together these data show that the RhopH complex can associate with the erythrocyte membrane and suggests that the soluble form can either insert into erythrocyte membranes or remain peripherally associated. It is difficult to distinguish between these two possibilities as previous studies showed that the members of the complex can be extracted using either detergent or bicarbonate buffer^{16,17}”

In several instances the authors use vague language that can lead to some confusion. Line 42-43. "...are deposited at the cell membrane of the infected cell and incorporated into the newly formed parasitophorous vacuole." Just stating that the proteins are released into the nascent PV would be clearer.

Author response: Following the reviewer's advice, we have changed this to"

"Here we show that the RhopH complex has not yet formed during merozoite invasion. At that time, Clag3 is directly released into the host cell cytoplasm, whilst RhopH2 and RhopH3 are released into the nascent parasitophorous vacuole."

We have also changed the Figure legend 6:

"Figure 6. Model for RhopH proteins trafficking onto the RBC surface. (a) RhopH2 and RhopH3 colocalize in late schizonts (1 and 2). Upon egress (3) free merozoites attach to new red blood cells (4). During invasion, rhoptry content is discharged and RhopH2 and RhopH3 are released into the nascent parasitophorous vacuole while Clag3 is predominantly injected into the host cell cytoplasm (5). Upon successful invasion, RhopH2 and RhopH3 are confined within the parasitophorous vacuole but show very little interaction (6). In trophozoites, RhopH2 and RhopH3 are exported via the PTEX translocon (7). In the host cell cytoplasm, both RhopH2 and RhopH3 associate with Clag3 and the RhopH complex consisting of RhopH2, RhopH3 and Clag3 can spontaneously insert into the erythrocyte membrane, where it forms the nutrient channel (8)."

Line 69: The different Clag proteins are best described as paralogues, rather than homologues (they may not have the same function; indeed, a different function has been proposed for Clag9 (although that has been disputed subsequently)).

Authors response: We have changed as per the reviewer's suggestion.

Line 123: The description of RhopH2 and RhopH3 as "cytosolic" is confusing. Do the authors mean that the proteins are free in the cytosol (as could be concluded from this description) or in vesicles in the cytosol (as the model in Figure 6 implies)?

Author response: We have clarified this according to the reviewer's suggestion:

‘In addition to the diffused signal, which could comprise soluble proteins, ER- or vesicle-inserted protein or a mix of all, well defined loci of RhopH2 and RhopH3 were also apparent. These were likely corresponding to the nascent rhoptries or rhoptry precursors (Figure 1a, b arrows).’

Line 130: ‘membrane-associated signal’ is somewhat non-specific, perhaps the authors can be more precise about which membrane.

Authors response: We appreciate this great suggestion but unfortunately due to limited resolution, we are unable to state whether the signal remains associated with parasite membrane or PVM. We have changed this to:

“Interestingly, RhopH3-mNeonGreen displayed an additional, signal in both trophozoite and schizont (Figure 1a, b, arrows). This signal colocalised with the membrane dye (Figure 1c,d) and disappeared just before parasite egress (Figure 1d, Suppl. Movie 2). This suggested the presence of mNeonGreen in the parasitophorous vacuole and but likely was not membrane associated as the limited resolution of light microscopy makes it difficult to distinguish.”

Lines 132 and 235: ‘parasitphorous’ should be ‘parasitophorous’.

Authors response: Thank you, we have corrected this.

Line 223: ‘were present predominantly at the parasitophorous vacuole’ sounds a bit confusing. One would expect the proteins to be in the parasitophorous vacuole or at the parasitophorous vacuole membrane.

Authors response: We have changed it to:

“In contrast, in ring-stage parasites, Clag3.1 was observed predominantly in the erythrocyte cytoplasm and it showed no colocalization with RhopH3, which was predominantly within the parasitophorous vacuole (Figure 4 b). Based on the data from invasion experiments, it is apparent that RhopH3 was on the inner surface of the parasitophorous vacuole membrane.”

Line 324:” ...the three proteins can spontaneously insert.” Presumably this is as a complex, it would be good to state this clearly.

Authors response: We have corrected this following the reviewer’s advice.

Line 338: Period missing at end of sentence.

Authors response: It has been corrected.

Line 378: Did the authors confirm integration by PCR (as written) or by PCR?

Authors response: The integration was confirmed by PCR. As the reaction was performed on genomic DNA, there was no need for reverse transcription. We thank the reviewer for correcting us. We have corrected this. In the text.

Line 465: Was the concentration of DDM used 2% or 2mM (as used in all other buffers)?

Authors response: It was 2% as stated in the text.

Line 489: 'washed' should be 'washing'

Authors response: We have corrected this.

Throughout: N-terminus, C-terminus should be N terminus, C terminus.

Authors response: We have now corrected this.

Line 122: The authors mention that the inability to obtain a Clag3-mNG fusion is likely owing to epigenetic switching and yet they are able to produce a fusion with the Flag tag. This is more compatible with a specific effect of mNG on the Clag3 fusion than epigenetic switching. Although not directly related to this manuscript, the authors may consider whether the ability to add short, loosely folded tags to Clag3, but not large, tightly folded tags, may indicate something about the mechanism through which the protein traverses the PVM.

Authors response: We thank the reviewer for this comment. A small tag like the HA or FLAG did not seem to pose any issues. We think that a bulky fluorescent tag might either lead the epigenetic switching or put the parasites expressing it at a disadvantage compared to those expressing the other epigenetic variant (for example due to a slower growth) and therefore with time we end up with parasites only expressing the non-fluorescent protein. We think that it was possible to obtain a fluorescently-tagged Clag3 by the authors of Ahmad et al because the parasite line the authors used has only one clag3 gene and therefore no switching can happen. Unfortunately, we can only speculate and thus we have changed the text to:

“We were not able to obtain fluorescently tagged Clag3.1 or Clag3.2, likely because the presence of a bulky tag interfered with the protein function and may have led to epigenetic switching between these two variants^{20, 32}.”

There are two very important findings that the authors should highlight more and address. First, the argument the RhopH2 and RhopH3 are transported separately is based on the finding that these two proteins are detected in different compartments (Supp Figure 3). This means that two rhoptry markers do not overlap and hence that neither is useful as a rhoptry marker. It would be useful to include a quantification of the number of instances in which the two proteins are not found together. Also, a colocalization of RhopH2 and RhopH3 with RAMA would indicate whether the compartments in which RhopH2 and RhopH3 reside really do represent rhoptries or rhoptry precursors. Also, the lack of colocalization of RhopH2 and RhopH3 is an important argument for the lack of complex formation in the parasites and should be added to the main text.

Authors response: We thank the reviewer for their insights. We acknowledge that we cannot certainly state whether the accumulation of RhopH proteins in late trophozoites and early schizonts corresponds to formed rhoptries or rhoptry precursors. These structures are very well distinguishable when performing super-resolution live imaging with fluorescent proteins but we fail to see well-defined structures upon fixation and antibody staining, most likely due to the limitations of the fixing protocol (such as antibody penetration and altered 3D architecture upon fixation). Obtaining reliable data would require creating more double-transfected lines with both RAMA and RhopH2 or RhopH3 fluorescently tagged. We therefore acknowledge this limitation in the in the main body of the manuscript and have now added a further discussion on the trafficking of RhopH2, RhopH3 into rhoptries:

“To determine if RhopH2 and RhopH3 are trafficked to rhoptries as a complex as previously suggested¹⁶, we used CRISP-Cas9 to create a parasite line with RhopH2 tagged with mNeonGreen and RhopH3 tagged with mScarlet. In late trophozoites, both proteins were detected in the cytoplasm, corresponding to the newly synthesised protein, as well as a defined rhoptry localisation. However, some of these early rhoptries showed only mScarlet or mNeonGreen signal (Supplementary Figure 3), consistent these two proteins being trafficked to the newly forming rhoptries or rhoptry precursors separately. This would agree with previous observations that depletion of either RhopH2 or RhopH3 does not affect rhoptry

localization of the other ¹⁶. However, another study has suggested that trafficking of Clag3 into the rhoptries may be somehow linked to that of RhopH2 and RhopH3 as this protein does not localise to the rhoptry in the absence of RhopH2 or RhopH3 ¹⁶. “

“

We have also included the quantification of the % of loci with a strong colocalization of both signals in Supplementary Figure 3b:

Second, the authors show that there is a delay in the export of RhopH2 and RhopH3, as these proteins are still in the PVM in the ring stage. This is a significant finding, as it suggests that export through the PTEX can be regulated. RESA, another PTEX cargo, is exported very soon after invasion, so something must be retaining RhopH2 and RhopH3. Costaining with RESA or another early exported protein (REX proteins, perhaps) would show that whereas these exported proteins can be exported, the RhopH proteins are held in the PV. The authors may have identified a new level of regulation of protein export, which should be highlighted more.

Authors response: We appreciate this insightful comment from the reviewer. We are not sure how the export of RhopH2 and RhopH3 from the PV is regulated. In fact, we think this export must start early on and is likely to happen gradually, as we can detect discrete RhopH2 and RhopH3 foci on the surface of the infected red blood cell even at the ring stage, when the parasites are still resistant to sorbitol treatment (Figure 3 a,b).

We think that the regulation of protein export and its timing deserves a separate study. However, we have now included a relevant entry in the discussion to address this valuable insight:

“Clag3 remains in the host cytoplasm until RhopH2 and RhopH3 are exported via the PTEX translocon, thus allowing the three proteins to associate and incorporate into the erythrocyte membrane. It remains to be determined how the timing of the nutrient channel assembly is regulated by the export of RhopH2 and RhopH3 from the parasitophorous vacuole. Given that discrete foci of RhopH2 or RhopH3 can be detected throughout the ring stage (Figure 1 a,b), the export must happen over time and be regulated, however, the susceptibility to sorbitol at a later stage could reflect the time when a critical mass of the protein has been exported allowing for enough functional channels to be assembled on the RBC surface.”

Reviewer #2 (Remarks to the Author): Pasternak et al. use sophisticated imaging technologies with transfected *P. falciparum* lines to study association and trafficking of RhopH proteins involved in formation of the parasite nutrient uptake channel, PSAC. Their findings suggest a revised model of how the proteins form a ternary complex and implicate incorporation into the host membrane to form PSAC as a last step in RhopH trafficking. The work is of a good quality and the interpretations reasonable. However, the revised model would benefit from stronger experimental evidence; discrepancies with prior studies require better discussion also.

Authors response: We would like to thank the reviewer for their positive feedback. We have addressed the specific comments as outlined below.

1. The most fundamental difference between the proposed model and prior studies is in how the RhopH complex transits from its manufacture to rhoptries and eventually to the cytoplasm of the next erythrocyte hours after invasion. Prior data suggests assembly of the ternary 1:1:1 complex during translation and a stable association that persists through trafficking to rhoptries and, at least, into the parasitophorous vacuole. In contrast, the authors appear to propose that the three proteins traffic to the rhoptry individually, that they may or may not associate within the rhoptry, and that during invasion that they dissociate to allow Clag3 deposit directly into host cell cytosol while RhopH2 and RhopH3 pass through the parasitophorous vacuole as monomers; the authors' model then presumes assembly (or reassembly) in host cell cytosol and spontaneous insertion into the host membrane. There are a few problems that require discussion in the paper (at a minimum) or new experiments to strengthen the evidence for the new model.

a. The authors acknowledge (Lines 117-118) that the “depletion of either RhopH2 or RhopH3 prevents the trafficking of Clag3 to the rhoptry (ref. #16).” If the proteins transit to the rhoptry individually, Clag3 delivery to rhoptries is not expected to be adversely affected by loss of the other subunits. The authors should discuss this point and list possible explanations to reconcile those findings with their own equally interesting data.

Author response: We thank the reviewer for their insightful comment. We have now included a possible explanation on what might be happening and reference ref.#16 as follows:

“In late trophozoites, both proteins were detected in the cytoplasm, corresponding to the newly synthesised protein, as well as a defined rhoptry localisation. However, some of these early rhoptries showed only mScarlet or mNeonGreen signal (Supplementary Figure 3), consistent these two proteins being trafficked to the newly forming rhoptries or rhoptry precursors separately. This would agree with previous observations that depletion of either RhopH2 or RhopH3 does not affect rhoptry localization of the other ¹⁶. However, another study has suggested that trafficking of Clag3 into the rhoptries may be somehow linked to that of RhopH2 and RhopH3 as this protein does not localise to the rhoptry in the absence of RhopH2 or RhopH3 ¹⁶.”

b. lines 293-295: “The biggest caveat of this model is that it does not explain the observation that RhopH3 but not RhopH2 or Clag3, has been implicated in parasite invasion (refs 16, 17, 41). This would require that the complex, believed to be very stable, may dissociate temporarily.” This is not necessarily correct. Certainly, a stable exposed domain on RhopH3 could provide the enzymatic activity or receptor-ligand interaction that facilitates invasion without dissociating from other RhopH subunits. One may reasonably speculate that there is a temporary dissociation but “require” seems much too strong.

Author response: We agree with the reviewer and removed the sentence mentioned above. We have also added an explanation on why the complex would dissociate at the translocation step, given that the translocation requires protein unfolding. The paragraph now reads as follows:

“This model does not explain the observation that RhopH3 but not RhopH2 or Clag3, has been implicated in parasite invasion ^{16, 17, 40}. There are also mixed reports regarding the role of the PTEX translocon in the export of RhopH complex from the parasitophorous vacuole. While all previous studies have found that export via the PTEX translocon was required for RhopH2 and RhopH3 to reach the host cell, conflicting results have been reported regarding Clag3^{16, 31}. Furthermore, since translocation via PTEX requires protein unfolding, the complex would dissociate at this step. Finally, the complex would be trafficked in the parasite-infected erythrocyte cytoplasm and be incorporated into the erythrocyte membrane and how this occurs is unknown. Thus, if all components of the PSAC channel were already associated inside the PV, what mechanism prevents them from incorporation into the parasite membrane or parasitophorous vacuole membrane?”

c. lines 175-177 regarding association of the three subunits within rhoptries: “These results suggest the RhopH complex is either not formed at this stage, in contrast to previous suggestions or that the complex dissociates during invasion.” The association of Clag3 and RhopH2 within rhoptries has been supported by FRET studies (see Fig. 6 of PMID: 32900800). As that paper used similar fluorescent tagging of RhopH subunits and live-cell imaging to examine the same questions asked by the authors, it seems important to cite it and discuss points of agreement and disagreement.

Author response: Following the reviewer’s excellent suggestion, we have attempted FRET on the RhopH2-mNeonGreen + RhopH3-mScarlet line, following similar strategy as by Ahmad et al (photobleaching the acceptor). However, as shown below, we did not detect any FRET in the parasite rhoptry. This could be because the distance between fluorescent tags on both proteins is larger than 10 nm and therefore no energy transfer can occur.

RhopH2-mNeonGreen + RhopH3-mScarlet

The distance could be too large for energy transfer to occur even if a complex between RhopH2 and RhopH3 was formed. Therefore, it was important to attempt FRET in the region where both proteins are known to form a complex – on the membrane of the infected red blood cell at the trophozoite stage, when the nutrient channel is formed. Unfortunately, because of the processing of the C terminus of RhopH3 (ref #16), the fluorescent tag is cleaved off before RhopH3 reaches the surface of the infected RBC. We therefore attempted to detect FRET on immunostained parasite lines, where RhopH2-HA was labelled with a monoclonal anti-HA antibody and a secondary antibody conjugated with Alexa-488 while RhopH3 was labelled with a polyclonal antibody and a secondary antibody conjugated to Alexa-594. Unfortunately, no FRET was detected on the RBC membrane, despite clear

antibody labelling (shown below). This suggests that the distance between the fluorophores (likely large due to the indirect antibody labelling) was too large to detect energy transfer and we were not able to use this technique to reliably answer the reviewer's questions.

We now include these experiments in the Supplementary Material and discuss the study by Ahmad et al in the manuscript:

In the Discussion:

“Recent FRET studies have also shown that RhopH2 and Clag3 remain in close proximity inside the rhoptries and on red blood cell surface, but there is no data on their association during and immediately after the merozoite invasion⁴³.”

And further:

“Firstly, our results indicate that RhopH2 and RhopH3 are trafficked to the rhoptries independently. This is consistent with previous observations that the absence of either RhopH2 or RhopH3 has no impact on the rhoptry localisation of the other¹⁶. On the other hand, trafficking of Clag3 into rhoptries might be linked to the trafficking of RhopH2 and RhopH3 as Clag3 fails to localise to the rhoptry in the absence of RhopH2 or RhopH3¹⁶. Also based on FRET studies, RhopH2 and Clag3 remain in close proximity inside the rhoptry⁴³ but the nature of this association remains to be determined, in particular whether they form a stable protein complex given they remain associated in pulldown experiments.”

d. The co-IP experiments in Fig. 4d suggest a model where the ternary complex between Clag3, RhopH2, and RhopH3 is formed in the rhoptry, with the proteins transiting to this organelle separately. Then, upon egress and invasion, they rapidly dissociate to allow Clag3 secretion into the host cell cytosol very early and RhopH2 and RhopH3 deposition into the parasitophorous vacuole. After these proteins cross the PVM, presumably via PTEX, the three proteins reassociate to allow membrane insertion. The authors' imaging is high-quality, and so, compellingly draws one to this model. At the same time, the RhopH complex structure revealed large surface areas of interaction between Clag3-RhopH2 and Clag3-RhopH3; the ternary complex was also reported to be quite thermostable (ref. 16). There is little or no precedent for such a stable complex to be rapidly disassembled. Given the

complex sites of interaction between the member proteins in the RhopH cryo-EM structure, it also seems hard to accept that they would come back together quickly, as invoked by the authors to account for an only modest reduction in Clag3 pulldown by RhopH2 at the ring stage: “Despite this, significant amounts of Clag3 were detected. This may be a result of the association of proteins as they are solubilised and membrane compartments and physical barriers removed allowing them to interact with each other.” Can the authors design an experiment to support or refute this concern? At the least, the Discussion section could include the authors’ thoughts on this conundrum.

Author response: We thank the reviewer for this insight and absolutely agree that it is unlikely that a formed complex would dissociate and reform. We have amended any results and discussion on this to make it clear and have changed the passage in the Results section to reflect another possibility that a fraction of Clag3 is also released into the PV space. The passage reads now:

“Despite this, significant amounts of Clag3 were detected. This likely results from the association of proteins after solubilisation when membrane compartments and physical barriers are removed allowing them to interact with each other. Surprisingly, the association between RhopH2 and RhopH3 was greatly reduced directly after invasion (Figure 4 d), despite both proteins remaining in the parasitophorous vacuole (Figure 4 a). This was consistent with our observation that RhopH2 and RhopH3 have very little overlap in their subcellular localisation when they are discharged into the host erythrocyte membrane during invasion (Figure 3). The recent Cryo-EM structure of the soluble form of RhopH complex^{23, 24} indicates that Clag3 is the core component while RhopH2 and RhopH3 remained associated with the complex via interactions at the opposite ends of Clag3. This agrees with our observation that in the absence of Clag3, the association between RhopH2 and RhopH3 does not occur. We have also attempted to detect protein association using FRET but did not observe any energy transfer in our experimental setup (Supplementary Figure 7).”

2. Figure 5 and interpretation of hemolysis induced by addition of RhopH complexes to erythrocytes or ghosts.

a. lines 269-271. Although not explicitly stated, the authors appear to interpret hemolysis as incorporation of RhopH and formation of PSAC that increases uninfected erythrocyte/ghost

osmotic fragility. I would not expect PSAC induction to produce hemolysis in an overnight PBS incubation as used by the authors because PSAC has a remarkably low Na⁺ permeability (see PMID: 14563534; Na⁺ is the primary osmotically active solute in PBS). I might consider alternate explanations for the observed and still interesting hemolysis. This experiment is also not described in the materials and methods, making the experiment more difficult to judge.

Author response: We thank the reviewer for this very useful comment. We acknowledge that we did not explain the interpretation of the RBC lysis properly. We agree with the reviewer that the release of haemoglobin reflected non-physiological interaction with the RBC membrane. We have added the following explanation into the main text:

“The lysis leading to the release of haemoglobin was unexpected and we speculate that could be due to the abnormal interaction with the external surface of the red blood cell since physiologically the complex would interact with the inner membrane leaflet on the cytoplasmic side. The insertion attempt from the outer leaflet side, without the presence of supporting cytoskeleton elements, could possibly result in membrane permeabilization and lead to the release of haemoglobin.”

In fact, the permeabilization of the RBC membrane has been confirmed with a fluorescent 2-NBD-glucose, where some RBCs appear to be have an empty, ghost-like appearance (shown below):

Control

2-NBD-glucose

+RhopH

2-NBD-glucose

As requested by the reviewer, we have also added the relevant method section:

“Red blood cell lysis

For the erythrocyte lysis experiment, 5 μ l of purified protein sample (0.4 μ g / μ l) diluted 1:10-1:20 in PBS, bovine albumin was mixed with 50 μ l of red blood cells (50% haematocrit) and incubated for 10 min at RT, then 600 μ l of PBS was added and samples were incubated at 37°C overnight. The control samples included bovine albumin at the same concentration as the RhopH complex, purified RhopH complex that had previously been heat-inactivated at 95°C for 10 min or the buffer in which RhopH complex was stored. Following the overnight incubations, samples were centrifuged at 13,000 g, supernatant was collected the absorbance was measured at 450 nm. “

b. Fig. 5c is labeled “Membrane incorporation”. From the Methods, I understand the authors washed the pellet twice with PBS. This is insufficient to remove weakly adherent proteins that are not “incorporated” into the membrane. Association with an ultracentrifugation pellet despite alkaline extraction (e.g. 100 mM NaCO₃ treatment for 30 min as used in ref. 16) is generally required to confidently implicate insertion into membranes.

Author response: The reviewer makes a valid point. Unfortunately, based on previous experiments, which show that the RhopH complex can be extracted by either detergent or bicarbonate buffer, it is difficult to distinguish between the full membrane incorporation vs membrane association of the RhopH complex. Daisuke et al, 2017, showed that all of RhopH3 and a substantial proportion of RhopH2 and Clag3 can be extracted using bicarbonate buffer. Counihan et al, 2017 managed to extract majority of RhopH2 using bicarbonate buffer as evident from very little protein left in the pellet. It has also been shown that bicarbonate buffer can be used to extract integral proteins with moderately hydrophobic transmembrane domains (Kim H, et al, 2015 Protein Sci. 2015, PMC4815233). Together, this makes it very difficult to use this method to assess whether the in vitro incorporation resulted in a membrane integration or association.

Counihan et al, 2017 Figure 2

Daisuke et al, 2017, Figure 1

We also believe that the protein presence in the pellets washed with PBS was not due to unspecific sticking of the protein to membranes as we repeated the same experiment with synthetic liposomes with the lipid composition similar to that of RBC membrane, and observed very no RhopH3 or Clag3 association with the liposome pellets (shown below).

Therefore, we believe the signal described in Figure 5 was specific although we can't distinguish whether the complex was fully integrated or membrane associated. We therefore follow the reviewer's suggestion and discuss this caveat in the text:

“Together these data show that the RhopH complex can associate with the erythrocyte membrane and suggests that the soluble form can either insert into erythrocyte membranes or remain peripherally associated. It is difficult to distinguish between these two possibilities as previous studies showed that the members of the complex can be extracted using either

detergent or bicarbonate buffer ^{16, 17}. This insertion/association with erythrocyte membranes seemed specific, as no association was detected in synthetic liposomes (Supplementary Figure 6).”

3. lines 136-142: interpretation of Supplementary Fig. 2 images obtained with super-resolution live imaging. These images are lovely, but I would be more cautious about interpreting where the N- and C- fragments localize based on imaging with two different fluorescent proteins. These proteins have different emission efficiencies so it wouldn't be surprising that both give a signal in the rhoptry (where the protein is concentrated) but give differing signals on the membrane, where there is less protein. To more confidently support such a model, it is generally suggested to swap the two fluorophores (i.e. make a new transfectant that expresses NeonGreen-RhopH3-mScarlet), but this may be prohibitively difficult. This concern also applies to lines 148-150 and Supplementary Fig. 3, where differing intensities of the two reporters are interpreted as implicating separate trafficking of RhopH2 and RhopH3, possibly to separate rhoptries. Also relating to this data, I am skeptical that the c-terminus of RhopH3 associates with membranes (e.g. parasitophorous vacuolar membrane or erythrocyte membrane) without the N-terminus because prior work implicates proteolytic processing near the C-terminus (>10 kDa from end). If correct, this would yield a small C-terminal fragment that lacks a predicted TM. Prior studies suggest that this fragment is rapidly degraded or lost to spent media upon egress (ref. 16, Fig. 6B). N.B. Ref. 16, Fig. 6C and 6F also shows that the vast majority of RhopH3 is proteolytically processed before egress and reinvasion.

Author response: We thank the reviewer for these insights. We agree with the reviewer's comment that different emission efficiencies of different fluorescent proteins might lead to inaccurate conclusions when compared to each other. To avoid this, we actually looked at the distribution of signal intensity for mNeonGreen or mScarlet relative to the background signal and/or the signal in the cytoplasm. We therefore didn't directly compare signal intensities between mNeonGreen and mScarlet but rather a distribution of each signal. Furthermore, we actually did swap the fluorophores at the C terminus of RhopH3. While in Figure 1 as well as Supplementary Figure 1, RhopH3 N terminus is tagged with mNeonGreen, in Supplementary Figure 3 it is tagged with mScarlet. Both fluorophores show similar localisation pattern (i.e. rhoptry + membrane-associated signal). We agree with the reviewer that the lack of a TM at

the C terminus makes this fragment unlikely to be membrane-inserted. Within the limited resolution of light microscopy, it seems to colocalise with a membrane dye but we are not sure about the nature of any potential membrane interaction. As the signal disappears shortly before the egress (Figure 1 and Supplementary Figure 3), we think this small cleaved fragment might be within the parasitophorous vacuole, although the nature of its secretion into the PV would be unclear. This would, however, explain the previous observation that the C-terminal fragment is lost before the next cycle (ref. 16). This would explain the previous observation that the C-terminal fragment is lost before the next cycle (ref. 16). Despite this, the same study suggests that the unprocessed form still gets transferred into the newly invaded red blood cell although the C terminus is never exported from the PV and thus does not reach the surface of the RBC. This is consistent with our observations (Figure 2, mNeonGreen present during invasion; Figure 1 – no mNeonGreen signal observed on the RBC surface). We now clarify this in the main body:

“Interestingly, RhopH3-mNeonGreen displayed an additional, signal in both trophozoite and schizont (Figure 1a, b, arrows). This signal colocalised with the membrane dye (Figure 1c,d) and disappeared just before parasite egress (Figure 1d, Suppl. Movie 2). This suggested the presence of mNeonGreen in the parasitophorous vacuole and thus might not actually be membrane associated but only appears so due to the limited resolution of light microscopy.”

Less important points: lines 134-136: I could not find the details of how a double-tagged RhopH3 parasite line (N-terminal mScarlet and C-terminal mNeonGreen) was made. Sequential transfection vs. a single larger insert? Also, the precise position of the mScarlet relative to the signal sequence is important to describe. To this reviewer’s knowledge, this is the first successful N-tagging of any RhopH protein that preserves trafficking and function. As this is not trivial, I would encourage the authors to include a ribbon schematic and highlight this accomplishment. Fig. 2d implies that an N-terminal FLAG tag has also been added. Could the authors provide details for this line, and also RhopH2-HA, Clag3.1HA lines?

Author response: We thank the reviewer for their kind words and the suggestion to create a ribbon schematic of protein tagging. These are now included in the Supplementary Figure 1f as shown below:

Lines 121-123 “We were not able to obtain fluorescently tagged Clag3.1 or Clag3.2, likely due to epigenetic switching between these two variants.” But, the authors generated a Clag3.1HA parasite? There are also parasite lines with a single clag3 gene, termed clag3h (PMID: 1815530), which have successfully been used to tag clag3 with fluorescent proteins.

Author response: We thank the reviewer for this comment. A small tag like the HA or FLAG did not seem to pose any issues. We think that a bulky fluorescent tag might either lead to the epigenetic switching or put the parasites expressing it at a disadvantage compared to those expressing the other epigenetic variant (for example due to a slower growth) and therefore with time we end up with parasites only expressing the non-fluorescent protein. We think that it was possible to obtain a fluorescently-tagged Clag3 by the authors Ahmad et al (PMID: 1815530) because the parasite line the authors used has only one clag3 gene and therefore no switching can happen. Unfortunately, we can only speculate and thus we have changed the text to:

“We were not able to obtain fluorescently tagged Clag3.1 or Clag3.2, likely because the presence of a bulky tag interfered with the protein function and may have led to epigenetic switching between these two variants^{20, 32}.”

Lines 160-161 “fixed at 1 min, 10 min and 30 min post invasion” implies greater certainty about the timing of invasion and fixation for individual cells in a population than I believe is possible because even tightly synchronized cultures have a distribution of parasite ages. Am I missing something here? The use of RON4 to define progression is elegant and sufficient.

Author response: The reviewer is absolutely right that even tightly synchronised parasites still display a range of ages. We have dramatically reduced this problem by the way our invasion assay was performed: synchronised and purified schizonts were further incubated in E64 to inhibit egress, allowing younger parasites to develop. Free merozoites were released by passing mature schizonts through a 1.2 µm filter and mixed with red blood cells as described in the Materials and Methods section. Fixing at two different time points: 1 min 30 s and 10 min allowed us to capture all stages of invasion: predominantly early to mid-stage invasion in the first time point, and a mix of all stages but predominantly late or completed invasions at the 10 min mark. As the reviewer noticed, the use of RON4 allowed us to further distinguish at which stage of invasion we look at.

Reviewer #3 (Remarks to the Author):

I think these are excellent findings but not yet definitive enough as the proximity of the postulated associations should and could be demonstrated using FRET. The authors have (some of) the relevant lines already and the choice of mscarlet and Neon Green seems to indicate they were thinking of the approach. The proteins are highly expressed so sensitivity should not be an issue. I'm a little mystified as to why they haven't reported attempting this.

The work is a sophisticated study of the assembly of the heterotrimeric facultative transporter complex consisting of one of the two Clag3 isoforms (RhopH1) and RhopH proteins 3 (H3)

and 2 (H2). This manuscript attempts, and to some extent succeeds, to deconvolute the complicated biology of the complex and track how the three members of the complex play merozoite associated roles and how they assemble into a 1:1:1 ratio heterotrimer that is inserted into the host red cell membrane to facilitate nutrient acquisition during parasite vegetative growth in the blood stream. They compare their data largely with that from a recent study of the Desai group (e-Life 10:e65282, although the Ho BioRxiv manuscript is also referred to) which followed the fate of Clag3 in detail and refine the model of associations, their timing and nature. The data are not unequivocal and the authors carefully explain the limitations however, with the tools in hand and generation of others more definitive data could be generated. CLAG 3 tagging was not achieved possibly due to switching between the pair of analogues – the Desai group deleted one of the pair of genes to allow tagging, an approach that would have improved this study.

Author response: We thank the reviewer for their constructive feedback. We regret we did not manage to achieve tagging of either Clag3.1 or Clag3.2 with fluorescent proteins. We did, however, manage to tag Clag3.1 with small epitope tags, which we then used for immunofluorescence, WB and protein pulldowns. A small tag like the HA or FLAG did not seem to pose any issues. We think that a bulky fluorescent tag might either led lead the epigenetic switching or put the parasites expressing it at a disadvantage compared to those expressing the other epigenetic variant (for example due to a slower growth) and therefore with time we end up with parasites only expressing the non-fluorescent protein. We think that it was possible to obtain a fluorescently-tagged Clag3 by the authors of Ahmad et al (PMID: 1815530) because the parasite line the authors used has only one clag3 gene and therefore no switching can happen. We have highlighted this in the text now:

“We were not able to obtain fluorescently tagged Clag3.1 or Clag3.2, likely because the presence of a bulky tag interfered with the protein function and may have led to epigenetic switching between these two variants^{20, 32}.”

For example, colocalization via FRET with H3 or H2 in the rhoptries of the segmented merozoite. According to the structure proposed by Desai and colleagues the proximity of the H2 and H3 C termini would be sufficiently close in the assembled complex to allow the appropriate energy transfer. Therefore, a very sensitive and unequivocal demonstration of

proximity/assembly of a subunit arrangement that could be anticipated is possible. The N/CT tagging experiment of H3. What has happened to the processed signal in mature schizonts immediately prior to egress? Contrast Fig1B with Supp Fig 2.

Author response: We thank the reviewer for pointing this out. The lack of a TM at the C terminus makes this fragment unlikely to be membrane-inserted. Within the limited resolution of light microscopy, it seems to colocalise with a membrane dye but we are not sure about the nature of any potential membrane interaction. As the signal disappears shortly before the egress (Figure 1 and Supplementary Figure 3), we think this small cleaved fragment might be within the parasitophorous vacuole, although the nature of its secretion into the PV would be unclear. This would, however, explain the previous observation that the C-terminal fragment is lost before the next cycle (ref. 16). Despite this, the same study suggests that the unprocessed form still gets transferred into the newly invaded red blood cell although the C terminus is never exported from the PV and thus does not reach the surface of the RBC. This is consistent with our observations (Figure 2, mNeonGreen present during invasion; Figure 1 – no mNeonGreen signal observed on the RBC surface). We now briefly discuss this in the main body:

“Interestingly, RhopH3-mNeonGreen displayed an additional, signal in both trophozoite and schizont (Figure 1a, b, arrows). This signal colocalised with the membrane dye (Figure 1c,d) and disappeared just before parasite egress (Figure 1d, Suppl. Movie 2). This suggested the presence of mNeonGreen in the parasitophorous vacuole and thus might not actually be membrane associated but only appears so due to the limited resolution of light microscopy.”

The dual tagging approach ought to allow a FRET based approach which might also discriminate between fragment colocalization and single molecule signals, i.e. there should not be complete overlap in the FRET signal emitted by mScarlet if the processed NT fragment is there on its own.

Author response: Following the reviewer’s comment, we have attempted FRET on parasite line with the double-tagged RhopH3. Given that both mNeonGreen and mScarlet signals were observed in the rhoptry, we tested if we could detect any FRET there. To this end, we bleached the acceptor (mScarlet), so that the lack of energy transfer would lead to an increase in the donor signal (mNeonGreen). Unfortunately, we were unable to detect any FRET, as

shown below (it comprises both the rhoptry and the “membrane-associated” signal. This could potentially be due to the distance between these two tags, but we acknowledge that the lack of a positive FRET signal in any of the tested conditions could also mean that our setup for FRET was not optimal. With the absence of FRET between the N and C terminus of RhopH3 in the rhoptry, it is impossible to use this technique to follow the dissociation of these two fragments throughout the parasite life cycle and we based our conclusions on the presence of mScarlet and mNeonGreen signals as well as antibody stainings, as previously described in the manuscript.

Similarly, a FRET-based approach should show the proximity of H3 and H2 in the rhoptries

as described in Supp. Figure 3. FRET signals typically indicate a <10nm distance which would indicate that a complex has or has not been formed. This is of interest as the H3 and H2 signals diverge so markedly upon invasion (and apparently at different rates which is not remarked upon) it begs the question as to why they might be co-complexed in the rhoptries. .

This dual tagged line and FRET could also show the relative disposition in Fig 3a b of H2 and H3 during the initial phases of invasion.

Authors response: Following the reviewer's excellent suggestion, we have attempted FRET on the RhopH2-mNeonGreen + RhopH3-mScarlet line, following similar strategy as outlined above (photobleaching the acceptor). However, as shown below, we did not detect any FRET in the parasite rhoptry. This could be because the distance between fluorescent tags on both proteins is larger than 10 nm and therefore no energy transfer can occur.

RhopH2-mNeonGreen + RhopH3-mScarlet

The distance could be too large for energy transfer to occur even if a complex between RhopH2 and RhopH3 was formed. Therefore, it was important to attempt FRET in the region where both proteins are known to form a complex – on the membrane of the infected red blood cell at the trophozoite stage, when the nutrient channel is formed. Unfortunately, because of the processing of the C terminus of RhopH3 (ref #16), the fluorescent tag is cleaved off before RhopH3 reaches the surface of the infected RBC. We therefore attempted to detect FRET on immunostained parasite lines, where RhopH2-HA was labelled with a monoclonal anti-HA antibody and a secondary antibody conjugated with Alexa-488 while RhopH3 was labelled with a polyclonal antibody and a secondary antibody conjugated to Alexa-594. Unfortunately, no FRET was detected on the RBC membrane, despite clear

antibody labelling (shown below). This suggests that the distance between the fluorophores (likely large due to the indirect antibody labelling) was too large to detect energy transfer and we were not able to use this technique to reliably answer the reviewer's questions.

This would aid accurate conclusions to be drawn resulting from the data in the later images of this figure. Whilst it can't help with CLAG localization unless the specialized line is generated as mentioned above that probably isn't necessary for this final section of the figure as CLAG is so distinct. Nevertheless, an indication of colocalization and possible complex formation in the rhoptries is not beyond this study with a little more work. At the moment, it's a somewhat partial answer.

FRET-based approaches would then allow timing of colocalization and perhaps even ordering of assembly of the complex to be determined as elucidated in Figure 4. Such approaches would allow more definitive language to be employed and avoid the hedging necessarily used from the pull-down approach.

Author response: We thank the reviewer for their suggestion on utilising the parasite lines we had created for FRET experiments. We have attempted these and now included the results in the Supplementary Material. As outlined above, we tried two approaches: using fluorescently-tagged proteins and immunolabelling. Neither method allowed us to observe any FRET between the fluorophores. Unfortunately, such negative result is not conclusive since we did not detect any FRET at the membrane of the infected RBC, where RhopH proteins form the nutrient channel. We therefore expanded our discussion on the complex formation and included reference to the previous FRET studies by Ahmad et al, as outlined below:

In the Discussion:

“Recent FRET studies have also shown that RhopH2 and Clag3 remain in close proximity inside the rhoptries and on red blood cell surface, but there is no data on their association during and immediately after the merozoite invasion⁴³.”

And further:

“Firstly, our results indicate that RhopH2 and RhopH3 are trafficked to the rhoptries independently. This is consistent with previous observations that the absence of either RhopH2 or RhopH3 has no impact on the rhoptry localisation of the other¹⁶. On the other hand, trafficking of Clag3 into rhoptries might be linked to the trafficking of RhopH2 and RhopH3 as Clag3 fails to localise to the rhoptry in the absence of RhopH2 or RhopH3¹⁶. Also based on FRET studies, RhopH2 and Clag3 remain in close-proximity inside the rhoptry⁴³ but the nature of this association remains to be determined, in particular whether they form a stable protein complex given they remain associated in pulldown experiments.”

Minor comments

L132 “parasitphorous” spelling

L145 “CRISP” spelling

L235 “parsitophorous” spelling

Supp figure 1. “genmic” spelling in part a

L826. “do” should be “to”

L847 separate and correct “tolate”

Author response: We have corrected the above spelling mistakes.

REVIEWERS' COMMENTS:

Reviewer #1 (Remarks to the Author):

The authors have answered my queries well. I do wonder what the value is of Figure 5b (the lysis of the erythrocyte when incubated with purified proteins overnight) when the results surprises even the authors and there is no obvious explanation for the result based on the proposed function of the protein. It does not seem to add to our knowledge of the function of the protein.

Reviewer #2 (Remarks to the Author):

The revised manuscript is improved, but a few smaller issues remain and require correction.

Line 153-156: The Ito et al paper (Ref 16) does not report that "depletion of either RhopH2 or RhopH3 does not affect rhostry localization of the other" as stated by the authors. Ref. 16 instead states "While RhopH2 was still detected in the R3gImS knockdown, it did not fully colocalize with RAP1 but instead had a more diffuse distribution; this suggests that RhopH2 may be retained in the parasite endoplasmic reticulum if it fails to interact with RhopH3. By contrast, RhopH3 appeared to traffic normally in GlcN-treated R2gImS parasites, as indicated by its apical colocalization with RAP1". Thus, the previous observations suggest that RhopH3 traffics normally in isolation, but that both RhopH2 and CLAG3 fail to traffic to the rhostry correctly if RhopH3 is removed. CLAG3 also fails to traffic to the rhostry if RhopH2 is removed.

This same mis-statement is made on lines 331-333 regarding Ref. 16. This point is an important distinction between the authors' observations and several studies from the Desai lab (not only Ref 16, but a stable association between RhopH members in schizonts is supported by FRET studies in ref 43 and by the cryo-EM structure in Ref 23, where an intact complex was harvested from mature infected cells with gentle freeze thaw without detergent addition). I suggest that the authors revise both of these sentences and present their interpretations more cautiously given this important discrepancy with several previous studies. For example, line 331 "our results indicate" seems too strong; "suggest" here and elsewhere may be more prudent.

Lines 335-338: This reviewer finds this sentence confusing. With the studies in Refs. 16, 43 and with the two structure papers (refs 23 and 24) as well as the authors' own pull-down experiments, "but the nature of this association remains to be determined, in particular whether they form a stable protein complex given that they remain associated in pulldown experiments" is at least confusing, if not inappropriately dismissive of a lot of data that contradicts the authors' preferred model.

Figure 5c and lines 295-297: Given that the proteins may be peripherally associated with membranes rather than integrated, please change the label above the graphic on Fig. 5c to "membrane association" rather the too strongly suggestive "membrane incorporation"

Line 299 and statements made in the Response to Reviewers: Please note that carbonate (CO_3 with a -2 charge and a pKa of 10.25) is used for extraction of peripherally associated membrane proteins. The authors' use of "bicarbonate" (HCO_3 with a -1 charge and a pKa of 6.1) would be embarrassing in a publication because only carbonate can provide adequate buffering at pH 11, as required for alkaline extraction of proteins.

Line 148: CRISPR-Cas9

Line 150: synthesized

Line 341: "structure of the ... RhopH complex" should cite refs 23, 24, not 16.

Dear Editor

Many thanks for the extra reviewers comments which are much appreciated. These comments have been addressed fully as shown in the marked-up version.

We look forward to your final decision on our manuscript.

Kind regards

Alan Cowman